# Ghost admixture in eastern gorillas

Harvinder Pawar[1], Aigerim Rymbekova [2,3], Sebastian Cuadros-Espinoza[1], Xin Huang [2,3], Marc de Manuel[1], Tom van der Valk[4,5], Irene Lobon [1], Marina Alvarez-Estape[1], Marc Haber[6], Olga Dolgova[7], Sojung Han [1,2,3], Paula Esteller-Cucala [1], David Juan [1], Qasim Ayub [8,9], Ruben Bautista [8], Joanna L. Kelley [10], Omar E. Cornejo [10], Oscar Lao[1], Aida M. Andrés[11], Katerina Guschanski [12,13,14], Benard Ssebide[15], Mike Cranfield[16], Chris Tyler-Smith[8], Yali Xue[8], Javier Prado-Martinez[1,8], Tomas Marques-Bonet [1,17,18,19,20] ✉ & Martin Kuhlwilm [1,2,3,20] ✉

Archaic admixture has had a substantial impact on human evolution with multiple events across different clades, including from extinct hominins such as Neanderthals and Denisovans into modern humans. In great apes, archaic admixture has been identified in chimpanzees and bonobos but the possibility of such events has not been explored in other species. Here, we address this question using high-coverage whole-genome sequences from all four extant gorilla subspecies, including six newly sequenced eastern gorillas from previously unsampled geographic regions. Using approximate Bayesian computation with neural networks to model the demographic history of gorillas, we find a signature of admixture from an archaic 'ghost' lineage into the common ancestor of eastern gorillas but not western gorillas. We infer that up to 3% of the genome of these individuals is introgressed from an archaic lineage that diverged more than 3 million years ago from the common ancestor of all extant gorillas. This introgression event took place before the split of mountain and eastern lowland gorillas, probably more than 40 thousand years ago and may have influenced perception of bitter taste in eastern gorillas. When comparing the introgression landscapes of gorillas, humans and bonobos, we find a consistent depletion of introgressed fragments on the X chromosome across these species. However, depletion in protein-coding content is not detectable in eastern gorillas, possibly as a consequence of stronger genetic drift in this species.

Gorillas are a member of the great apes and form a sister clade to *Homo* (human) and *Pan* (chimpanzees and bonobos). Extant gorillas consist of four recognized subspecies, which cluster into two species, a western species of western lowland (*Gorilla gorilla gorilla*) and Cross River (*Gorilla gorilla diehli*) gorillas and an eastern species of eastern lowland (*Gorilla beringei graueri*) and mountain gorillas (*Gorilla beringei beringei*)[1]. All gorilla subspecies are either endangered or critically endangered under IUCN criteria[2–4].

The subspecies are distributed across western and eastern Africa in a non-continuous manner (Fig. 1a). The current geographic ranges of the different subspecies differ by size, continuity and ecology, impacting connectivity and population sizes[5]. Western lowland gorillas are endemic to a largely continuous range of considerable size, whereas the other subspecies have much more fragmented distributions[6]. Likewise, western lowland gorillas exhibit the highest genetic diversity of the subspecies[5,7,8], indicative of long-term high effective population sizes,

**Fig. 1 | Gorilla samples used in this study. a,** Present geographic distribution of eastern gorillas, with that of the four gorilla subspecies shown in the inset. The newly sequenced samples are given in black, numbers of previously sequenced eastern gorillas are given in colour. GBG, *Gorilla beringei graueri* (Eastern lowland gorilla, *n* = 9); GBB, *Gorilla beringei beringei* (Mountain gorilla, *n* = 12); GGG, *Gorilla gorilla gorilla* (Western lowland gorilla, *n* = 27); GGD, *Gorilla gorilla diehli* (Cross River gorilla, *n* = 1). Shape files for the distribution of gorilla subspecies were obtained from IUCN. **b,** PCA with PCs 1 and 2 shown. **c,** PCA with PCs 3 and 4 shown.

while eastern gorilla effective population sizes are smaller[9]. Mountain gorillas are currently isolated in two discrete areas, the Virunga Volcanoes Massif and the Bwindi Impenetrable National Park. The Bwindi National Park is located at a lower elevation than the Virunga Volcanoes and as such has warmer temperatures[10,11]. Previous studies of the demographic history of gorillas did not incorporate information from all subspecies and were not conclusive, especially about the divergence time between the eastern and western clade[9,12–15]. This might be due to gene flow from unsampled lineages, which is probably widespread but is often insufficiently considered in evolutionary studies[16,17]. While uncovering such hidden introgression events in gorillas is not possible from ancient DNA from fossil remains, as has been performed in humans[18], it is possible to address such questions using genomic data from present-day individuals[19–22].

To address this question, we use high-coverage whole-genome sequences of 28 western and 21 eastern gorillas. In addition to previously published genomes[7,8], we sequenced the genomes of five mountain gorillas from the Bwindi National Park and one eastern lowland gorilla from the isolated population of Mount Tshiaberimu. These new genomes contribute to a more complete representation of the genomic diversity of eastern gorillas. Using this expanded dataset, representing all four known gorilla subspecies, we explored the demographic history of gorillas and specifically the hypothesis of ghost introgression, defined as gene flow from an unsampled archaic lineage. Given its substantial impact in their sister taxa of *Pan* and *Homo* as well as many other species[18,21,22], such ghost introgression events may explain some of the uncertainties in previous demographic models for gorillas. Using an approximate Bayesian computation (ABC) approach, we find evidence for introgression from an extinct lineage into the common ancestor of eastern gorillas and characterize some of the functional consequences of this introgressed genetic material.

## Results

### Eastern gorillas form two population clusters

We newly sequenced six eastern gorillas to high coverage (on average, 28.6×). After reprocessing the sequencing data from previous studies (Methods), we obtained a dataset of 49 individuals, with 27 western lowland gorillas, one Cross River gorilla, 12 mountain gorillas and nine eastern lowland gorillas (Extended Data Table 1). We performed a principal component analysis (PCA; Methods) to ascertain whether the newly sequenced individuals cluster with individuals from the same subspecies. The first PC separates western and eastern gorillas, as previously observed, and the second PC separates mountain gorillas from eastern lowland gorillas (Fig. 1b). Since the new individual from the isolated Mount Tshiaberimu population clusters within the distribution of the other eastern lowland gorillas (Fig. 1b), this individual is, as expected, considered a representative of this subspecies. The third PC reflects population stratification within western lowland gorillas, whereas the fourth PC separates the eastern gorillas, with the two mountain gorilla populations from Virunga and Bwindi at the extremes (Fig. 1c), explaining 3.2% of the variance. This is well in agreement with previous studies[7,8].

### ABC modelling favours a ghost lineage in eastern gorillas

To infer a demographic model for the four extant gorilla subspecies, we used a neural-network based ABC modelling strategy using windowed summary statistics and extensive simulations (Methods; Extended Data Fig. 1), based on a previous implementation in the *Pan* clade[22]. A main improvement is the implementation of a broad range of informative summary statistics (Supplementary Table 5), as is common practice for ABC studies in modern and ancient humans[23].

We first established a demographic null model of the four populations (Extended Data Fig. 2a and Supplementary Fig. 5), on the basis of previous studies[3,7,8,24–27]. Notably, although none of these studies incorporated whole-genome data from all subspecies, our inferred parameters are largely coherent with previous work (Supplementary Tables 1 and 2). Nevertheless, unaccounted demographic events such as ancient population structure or ghost admixture could affect parameter estimates[16], particularly given evidence in other great apes[22]. Initial exploratory analyses with $f_4$-statistics and admixture graphs (Supplementary Section 2) did not show any asymmetries between the four gorilla terminal populations, which would arise if ghost admixture had occurred in any of the individual subspecies. However, this does not exclude the possibility of ghost admixture into the common ancestor of eastern or western gorillas, which these methods cannot assess. To account for this and explicitly test if ghost admixture could improve the inferred null demographic model (model A), we considered two

more complex demographic models, in which we added the possibility of 'ghost' introgression into the common ancestor of eastern gorillas (model B) and western gorillas (model C). We assessed the robustness of our ghost models B and C using a wider parameter space (Supplementary Figs. 8–10; Methods), resulting in coherent posteriors with those observed in models B and C (Supplementary Table 2), albeit with wider confidence intervals (CIs), as expected given the increased model complexity (Supplementary Fig. 8). We performed a formal comparison of these models (Methods), to determine which fits the empirical data best. Model B, with archaic gene flow to the common eastern ancestor had the highest posterior model probability of 0.9973, compared to models A (0.0027) and C (0) and a substantially higher Bayes factor (374 versus 0.0027 for model A and 0 for model C). In a cross-validation analysis, the model with archaic introgression into eastern gorillas was clearly distinguishable from the model without (Supplementary Table 3). We conclude that a model with archaic introgression into the common eastern ancestor best explains the observed summary statistics in the empirical data (for full posterior distributions see Extended Data Fig. 3).

We infer that eastern gorillas experienced bottlenecks and generally had lower effective population sizes than western gorillas, while mountain gorillas and eastern lowland gorillas experienced a particularly strong population decrease (Supplementary Tables 1 and 2), as described previously[8,24]. We infer that the eastern subspecies split at 15,000 years ago (ka) (14–16 ka, 95% credible interval (CrI), Supplementary Tables 1 and 2). In agreement with previous studies[13], we see a population expansion in western lowland gorillas ~40 ka. Our null demographic model infers a large ancestral population size for western gorillas (effective population size, $N_e$ = 98,135), in comparison to that of other gorilla populations, as well as a split between the western gorilla subspecies at ~454 ka (448–456 ka 95% CrI). Considering that not all summary statistics could be calculated for Cross River gorillas (where only one sample was available) and gene flow between western gorilla subspecies was not modelled, we caution that the confidence in this split time might be low. Finally, we infer that gorillas diverged into two species ~965 ka (729–1,104 ka 95% CrI), which is within the higher range of previous estimates[9,12,28].

For simplicity, we modelled extant admixture as single migration pulses over one generation, finding a small contribution of gene flow from the common eastern ancestor to the western lowland gorillas of 0.80% (0.06–2.14% 95% CrI), as well as from western lowland gorillas to the common eastern ancestor of 0.27% (0.22–0.43% 95% CrI). We infer a contribution of 2.47% of gene flow from an archaic source into the common ancestor of eastern gorillas, with a narrow 95% CrI of 2.38–2.49% (Fig. 2c). We infer that this ghost population diverged from the extant gorilla lineages ~3.4 million years ago (Ma) (2.98–3.8 Ma, 95% CrI). We estimate the timing of this ghost gene flow to have occurred 38,281 years ago, although the CrIs for this parameter are wide (22–108 ka, 95% CrI) (Fig. 2a,c). By contrast, the posterior distributions for the archaic introgression proportion and the gorilla–ghost divergence time have narrow CrIs, indicating a strong support with clear peaks for these parameters (Fig. 2c). In contrast, our ABC analysis of model C does not confidently infer a contribution of a deeply divergent external lineage into the common ancestor of western gorillas. Instead, the best fit of this model suggests a 0.17% (0.09–0.4%, 95% CrI) contribution from an external lineage at ~1.1 Ma into the common ancestor of all extant gorillas (Supplementary Table 2). This marginal contribution is inferred to originate from an external lineage which diverged from extant gorillas 1.9 Ma (1.5–3 Ma 95% CrI).

### The ghost introgression landscape in eastern gorillas
Having established that a model of ghost introgression into the common eastern ancestor provided the best fit to the empirical data, we aimed to identify the putative introgressed fragments in the genomes of eastern gorillas. To explore this landscape of ghost introgression,

we implemented two independent approaches: the $S^*$ statistic[19,20,29] and the SkovHMM method or hmmix[30]. The $S^*$ statistic detects highly divergent windows relative to an outgroup, under a given demographic model, as introgressed sites[19,20,29]. Hence, the $S^*$ approach depends on the availability of a demographic model. By contrast, hmmix does not rely on a demographic model to identify putatively introgressed regions but instead uses the density of private mutations in the ingroup to partition the genome into 'internal' and 'external' fractions, walking in small windows of 1,000 base pairs (bp) along the genomes[30]. Hence, although both $S^*$ and hmmix target the same signature of ghost introgression, the algorithms are distinct.

We simulated the expected null distribution of $S^*$ scores for eastern gorillas with posterior parameter estimates from model A, that is a model without ghost introgression. This yields insights into the presence of any outlier windows in our empirical data using the 99% confidence interval (CI) for expected $S^*$ scores, given the mutation density (number of segregating sites) in each window (Supplementary Fig. 11; Methods). Indeed, at this threshold we identify an excess of $S^*$ outlier windows, suggestive of introgression from an external source into the common ancestor of eastern gorillas: windows which fall outside the null expectation constitute, on average, 1.64% of eastern lowland genomes and 2.36% of mountain gorilla genomes, respectively (Supplementary Table 9).

We assessed the performance of the $S^*$ statistic using coalescent simulations where we could trace the introgressed fragments (Methods). The precision and recall are high, with a 90.96% detection rate of true introgressed fragments for eastern lowland gorillas (91.06% for mountain gorillas) at the 99% quantile (Fig. 2b, Extended Data Fig. 4 and Supplementary Table 7; Methods), comparable to the human–Neanderthal scenario[31]. Since the CrIs of the null demographic model encompass larger effective population sizes, which would lead to inflated rates of incomplete lineage sorting that might affect the expected distribution of $S^*$ scores, we also assessed how these parameters influence our findings. Using the maximum values within the 95% CrIs, we find that the recall of the $S^*$ statistic remains high, while the precision falls to 55.82% for eastern lowland gorillas (53.33% for mountain gorillas), reflecting an increase in the false discovery rate, as expected. We conclude that the $S^*$ statistic performed well in detecting introgressed fragments under our null model, even when assuming misspecification of the null model.

Analogous to previous work[22], we also used hmmix to detect introgressed windows[30], which performs well for the given demographic model (Fig. 2b), with precision and recall well above 80% (Supplementary Table 8). Considering the strong support for ghost admixture into eastern gorillas, we again used western lowland gorillas as the outgroup and eastern gorillas as the ingroup. We find that 1.48–2.97% of the individual eastern gorilla genomes are inferred as external at a strict threshold for the mean probability of 0.95, with an estimated introgression time of 37–41 ka.

While we observe sharing of the putative introgressed regions across the eastern species, sharing is higher within each subspecies, which again is more pronounced in the mountain gorillas than in the eastern lowland gorillas (Fig. 3a). This indicates that most of the putative introgressed regions are segregating rather than fixed. Pairwise nucleotide differences are elevated between eastern and western gorillas in putative introgressed regions in eastern gorillas, compared to random regions (Fig. 3b). Likewise, there is an excess of nucleotide differences between individuals of the eastern subspecies in the putative introgressed regions, indicative of an archaic origin of these regions (Fig. 3b).

The overlap of the autosomal hmmix fragments and the $S^*$ outliers within each individual is, on average, 42% for eastern lowland gorillas and 51% for mountain gorillas (Supplementary Table 11). For random genomic regions passing filtering criteria, the observed overlap is, on average, 6% for eastern lowland and 8% for mountain gorillas, suggesting

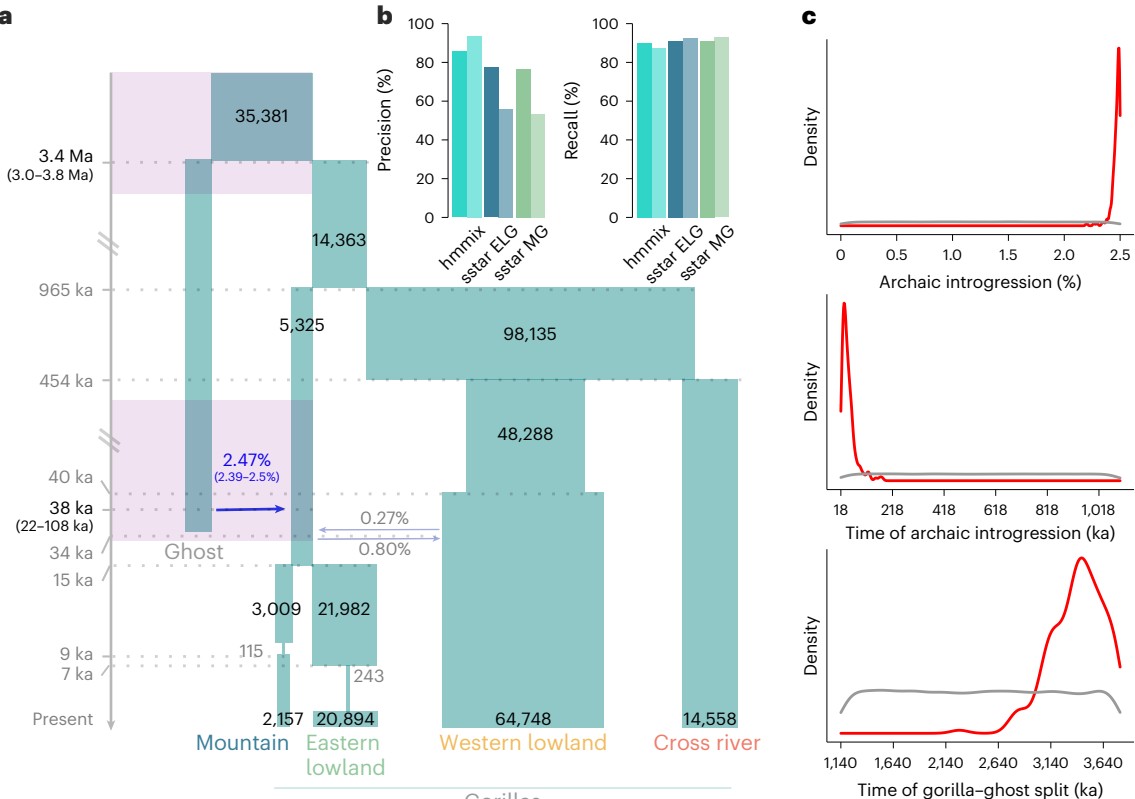

**Fig. 2 | ABC-based demographic model. a,** Model of gorilla population history with archaic admixture from an unsampled 'ghost' lineage into the common ancestor of eastern gorillas. The 95% CrI are shown for the archaic introgression proportion, timing of archaic introgression and archaic divergence (purple timeframes), inferred under ABC modelling. Numbers on blocks represent effective population sizes. **b,** Precision and recall of hmmix (at the 95% posterior probability cutoff) and $S^*$ (at the 99% quantile using sstar[31]) in simulated data using msprime. Precision (percentage of recovered introgressed fragments) and recall (percentage of true among inferred introgressed fragments) for hmmix and $S^*$ (for ELG, eastern lowland gorilla and MG, mountain gorilla). Dark bars represent performance using the model presented in **a**, light bars represent the 'worst' model with large $N_e$, in the case of hmmix to simulate the data to detect fragments, in the case of $S^*$ to obtain the expected distribution of $S^*$ scores. **c,** Posterior distributions for the archaic introgression proportion, time of archaic introgression and gorilla–ghost split time. The grey line indicates the prior distribution. The red line represents the posterior inferred with neural networks. Neural networks reduce the dimensionality of the summary statistics used and account for possible mismatch between the observed and simulated summary statistics[54].

that both methods detect to a large degree the same regions (Fig. 3c and Supplementary Table 14). We thus consider the regions in the intersect of the hmmix outliers and $S^*$ outliers as our high-confidence putative introgressed regions. The overlap between the two methods increases to 59% for eastern lowland and 68% for mountain gorillas, when using more lenient cutoffs for both methods, that is hmmix fragments of at least 40 kilobases (kb) and 95% CI $S^*$ outliers (Supplementary Table 12). Mountain gorillas (with the exception of Turimaso) consistently exhibit higher proportions of overlapping base pairs of the two methods than do the eastern lowland gorillas (Supplementary Tables 11 and 12).

### The interaction of selection and introgression

In contrast to archaic introgressed regions identified in humans and bonobos, the putative introgressed regions in eastern gorillas are not significantly depleted in genic content compared to random genomic regions (Fig. 3d). However, we find 127 megabases (Mb) of autosomal segments longer than 5 Mb that are depleted for introgressed fragments (Fig. 4). Further, we observe a signal of depletion in archaic fragments on the X chromosome (Fig. 3f), on a scale comparable to observations in modern humans[32] and bonobos[22]. The putative introgressed regions of eastern lowland gorillas exhibit a slightly higher proportion of likely deleterious sites than do mountain gorillas, as estimated by the GERP score (Fig. 3e). However, under alternate measures of mutational conservation (SIFT, PolyPhen-2 and LINSIGHT scores) the putative introgressed regions of both eastern gorilla subspecies follow random expectation (Supplementary Fig. 19). We also investigated the distribution of gorilla-defined regulatory element annotations from another study[33]. Here, across categories and populations, we only observe an excess of strong enhancers (sE) in mountain gorilla introgressed regions, compared to random regions (Supplementary Fig. 20). These are largely intragenic enhancers (Supplementary Fig. 21), which agrees with patterns of regulatory architecture observed in primate sE[33].

Introgressed fragments can carry beneficial alleles and to explore signatures of adaptive introgression within eastern gorillas we applied the method VolcanoFinder[34]. VolcanoFinder scans the genome for a signal of a distorted local site frequency spectrum consistent with a selective sweep surrounding an introgressed allele. Outliers of the VolcanoFinder approach (95% composite likelihood ratio) within the putative introgressed regions identified above were considered putative targets of adaptive introgression. We identify seven candidate regions for adaptive introgression (Extended Data Table 2), three of which are shared between eastern lowland and mountain gorillas. The region with the highest likelihood ratio (LR) in VolcanoFinder (chr. 12: 11090005–11324172; maximum LR = 246.2) contains the bitter taste receptor *TAS2R14*, within which we find several protein-coding changes (Supplementary Table 15).

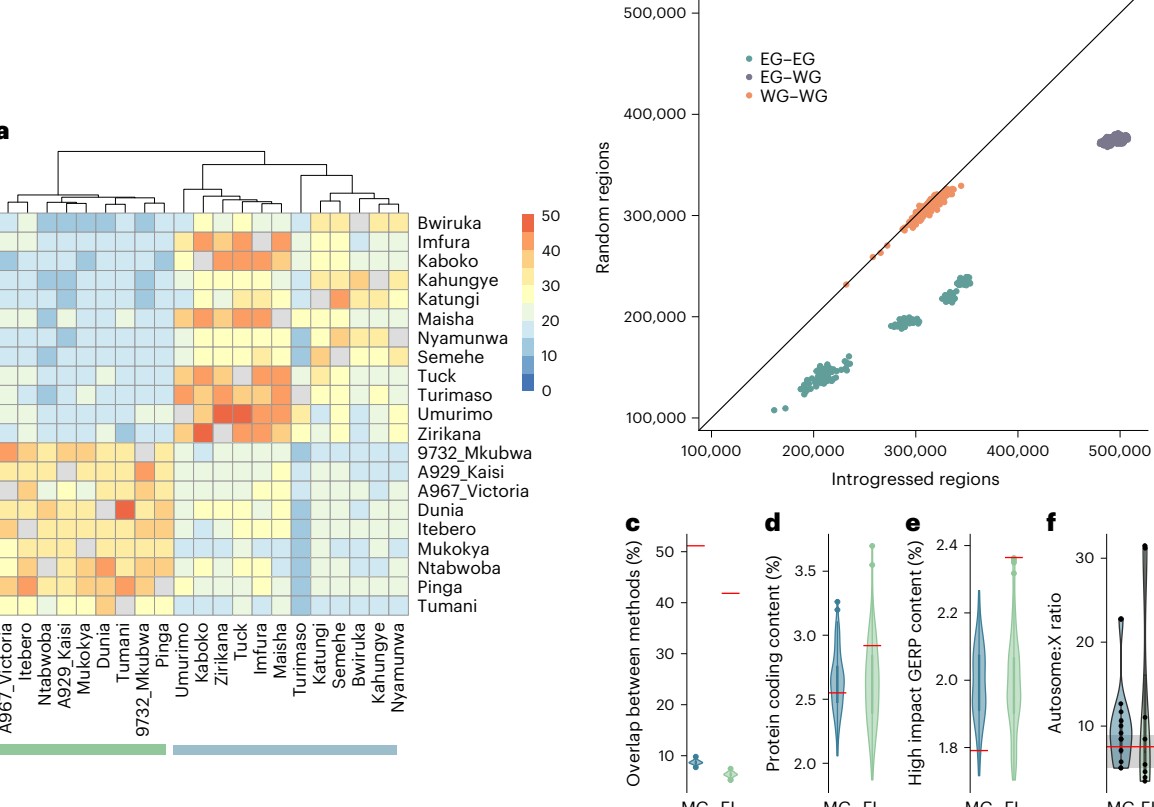

**Fig. 3 | Characterization of introgressed fragments. a**, Sharing of putative introgressed regions across eastern gorillas for autosomal regions detected using the $S^*$ statistic and hmmix. **b**, Pairwise nucleotide differences in introgressed regions (x axis) and in random regions (y axis) matched for length and proportion of positions with sufficient coverage (avoiding genomic regions without callable sites). Colours indicate the comparison: among eastern gorillas (EG–EG, green), among western gorillas (WG–WG, orange) and between eastern and western gorillas (EG–WG, purple). **c**, Percentage of overlapping base pairs in introgressed regions (red lines) and random regions (violin plots) for eastern gorillas. For details of the definition of random regions see Methods. **d**, Percentage of protein-coding content detected in introgressed regions (red lines) and random regions (violin plots) for eastern gorillas. **e**, Percentage of high impact GERP content detected in introgressed regions (red lines) and random regions (violin plots) for eastern gorillas. **f**, Autosome: X ratio of introgressed fragments inferred using hmmix for eastern gorillas (violin plots), with reference lines for the equivalent values for bonobos (red line) and humans (distribution as grey bar). In **c**–**f**: MG, mountain gorillas; EL, eastern lowlands. In **c**–**f**, data are presented in violin plots with overlaid boxplots, which represent the median and interquartile range (25th and 75th percentiles). In **f**, individual datapoints are additionally plotted as black circles. For **c**–**e**, the data in violin plots consist of population-wise means for n = 100 iterations of random genomic regions; for **f**, the data consist of hmmix fragments for n = 12 mountain gorillas and n = 9 eastern lowland gorillas.

## Discussion

Here, we present a demographic model inferred from representatives of all four extant gorilla subspecies, leveraging the most comprehensive dataset of gorilla genomes available to date and an improved estimate for gorilla mutation rate from extended pedigree data[35]. The newly sequenced whole genomes of mountain gorillas from Bwindi National Park are genetically close to those from Virunga but form a distinct cluster within their subspecies (Fig. 1b,c), confirming earlier results from microsatellite data[36]. Eastern lowland gorillas, as represented in our dataset, seem to form a genetically less differentiated population, which includes the individual from Mount Tshiaberimu. Nonetheless, sample size remains a limitation, as high-quality invasive samples are highly restricted for endangered species, given ethical and logistical constraints. A more fine-grained analysis of the evolutionary history and population structure of gorillas necessitates denser sampling, which most likely will only be possible through advances in the use of non-invasive samples. For example, a reconstruction of recent patterns of connectivity has been demonstrated from a large panel of faecal samples from chimpanzees[37]. Furthermore, considering the rapid decline of great ape populations over the past centuries, more

temporal sampling from historical specimens[24,25] has the potential to be highly informative on variation lost over time.

Previous estimates of demographic parameters varied greatly under different models, methods and input data[9,13–15]. The ABC approach presented here leverages population-wise summary statistics. However, since high-coverage, population-level whole genomes are not currently available for Cross River gorillas, a subset of the statistics could not be obtained for this subspecies (Methods; Supplementary Table 5) and those calculated may be relatively less informative (for example, number of segregating sites). For all other populations, multiple individuals were included, yielding a better representation of their diversity in the summary statistics. As such, we have lower confidence in parameters involving Cross River gorillas, such as the relatively large divergence time inferred for the western lowland–Cross River split. This divergence time represents 47% of the inferred eastern–western species split, compared to 26% estimated in a previous study which also inferred a more recent species split time[13]. We note that this difference may be attributed to our inclusion of more western lowlands gorillas, known to have high levels of population structure[9,38,39]. We also do not include gene flow between western lowlands and Cross

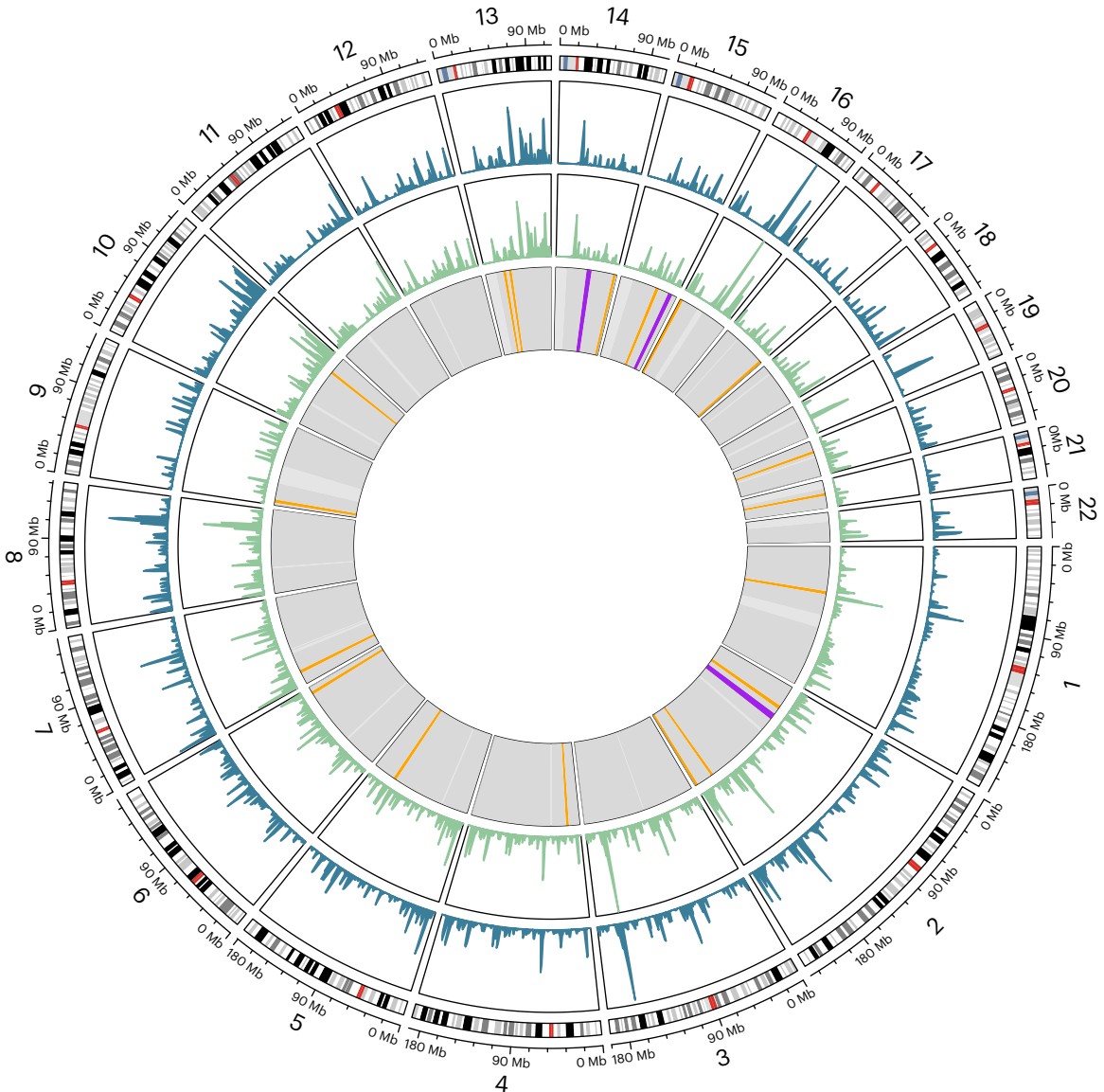

**Fig. 4 | Distribution of introgressed fragments.** Outer circle: karyogram of the autosomes based on the human genome (hg19). Second circle from outside: introgression landscape in mountain gorillas (blue), as cumulative amount of introgressed material in sliding windows of 2 million base pairs, Mb). Third circle from outside: introgression landscape in eastern lowland gorillas (green) in sliding windows of 2 Mb. Inner circle: long regions depleted of introgression content are shown in orange (length ≥5 Mb) and purple (length ≥8 Mb). Grey: genomic regions with sufficient data (>20% of 40 kb windows passing threshold). White: genomic regions without sufficient data.

River gorillas as a parameter in our modelling, which would reduce divergence estimates.

The inferred deep divergence time between the two species is at the upper end of previous estimates and conservative for the detection of putatively introgressed windows under the null model, since larger $S^*$ scores would be expected to result from an increased number of segregating sites[22]. Indeed, even approximate demographic models with large divergence times may allow a detection of external gene flow into a target population[31]. We demonstrate that the $S^*$ statistic performed well in detecting introgression under the null model inferred herein (model A), even if the true demography was deviating in terms of ancestral effective population sizes. Demographic modelling presented here finds the best model for gorilla demography to include archaic introgression from an unsampled 'ghost' lineage into the common ancestor of eastern gorillas. This accords with a growing literature on the prevalence of introgression from extinct lineages in humans[21,40],

bonobos[22] and other species[41,42], as well as theoretical predictions and simulations showing an impact of admixture from unsampled lineages that is probably common rather than exceptional[16,17]. Using extensive simulations, we find strong support for a model including archaic admixture into eastern gorillas, compared to a null model without such ghost admixture or a model of such an event in western gorillas. The latter may be rather considered similar to a model of deep substructure within gorillas, given the shallower times and small amounts of external gene flow inferred. However, we note that further ghost introgression events may exist beyond what we describe, for example with regards to much smaller amounts of ghost admixture into gorillas or with shallower divergence times of the ghost lineages or in the context of larger effective population sizes in western gorillas.

Our inference of 2.47% of ghost introgression is associated with high confidence as the posterior distribution is well differentiated from that of the prior (Fig. 2c). This estimate agrees well with the estimates

of genome-wide introgression proportions per individual inferred using the $S^*$ statistic and hmmix (Supplementary Table 9). We probably underestimate the timing of archaic introgression, since shorter introgressed fragments are more likely to be missed and another potential complication might be relatively high levels of homozygosity in eastern gorillas[8], leading to increased haplotype lengths. Our definition of putative introgressed regions as the overlap of outliers inferred with both the $S^*$ and hmmix methods (Fig. 3c) is conservative and on the order expected for these methods, considering their relatively high false-positive rates[31]. Nonetheless, these methods are currently the only reliable tools available for detecting introgressed fragments in comparably small datasets of non-human species, without the availability of a source genome[31].

A higher degree of sharing of putative introgressed fragments is observed among mountain gorillas than in eastern lowlands (Fig. 3a). This is consistent with smaller effective population sizes of these populations, increasing the impact of drift on introgressed genetic variation[18]. High levels of genetic drift and reduced efficacy of natural selection probably also explain the absence of a detectable depletion of genic content in introgressed regions, in contrast to observations in introgressed material of humans and bonobos. Likewise, we do not observe a coherent signature in mutational tolerance in gorilla introgressed material across different metrics, possibly due to genetic drift. Despite this, we do find some 'introgression deserts', that is regions depleted of introgressed material in the population (Fig. 4), possibly as a result of purifying selection[18] shortly after the introgression took place. Furthermore, we observe a reduction of introgression on the X chromosome, as also seen in humans and bonobos[22,30,32]. This is probably a result of strong purifying selection against introgressed variation, as seen in humans and other species[18,30,43], possibly as a result of a combination with multiple factors[44]. Biased dispersal patterns[45] and high reproductive skew of gorilla males[46] might have led to a further reduction of the male-haploid X chromosome in introgressed material. Even though the observed patterns are probably a combination of these factors, we can currently not discern their respective contributions.

We note that our definition of adaptive introgressed targets is highly conservative, as the intersection of the outliers of three different methods $S^*$, hmmix and VolcanoFinder as putative adaptive introgressed targets. However, in being conservative we aim to minimize the impact of potential false positives, which is a known caveat of the VolcanoFinder method[34,47]. However, at present this is the only method available to localize signatures of adaptive introgression without a source genome. Interestingly, three candidate genes contain putative functional variants segregating in eastern gorillas and fixed ancestral in western gorillas. One of these genes is *TAS2R14*, which encodes a taste receptor implicated in perception of bitter tastes[48] and contains six missense variants. Eastern gorillas typically have more herbaceous diets than the frugivorous western gorillas[11], as such taste receptors are plausible targets of adaptive introgression in eastern gorillas. Bitter taste receptors have been suggested as targets of recent positive selection in western lowland gorillas as well, including a region encompassing *TAS2R14* (ref. 13). It is possible that different mutations in the same region have been under selection in the different species. This could be interpreted in terms of the essential role of taste receptors to avoid toxicity. The gene *SEMA5A* contains a missense variant and a splice region variant; this gene has been associated with neural development, with implications in autism spectrum disorder[49]. However, the functional impact of these variants in gorillas demands further work in the future. Here, we do not find a contribution of adaptive introgression to altitude adaptation, a phenomenon observed in humans and other species[18,50]. In mountain gorillas and eastern lowland gorillas at high altitude, this adaptation is probably driven by different mechanisms, such as the oral microbiome[51].

In conclusion, our work contributes improved resolution to our understanding of the evolutionary history of eastern gorillas. Across individuals, we recover a putative 16.4% of the autosomal genome of an extinct lineage (Supplementary Table 13), adding to a growing literature revealing unsampled, now extinct lineages via analysis of variation present in present-day individuals.

## Methods

### Samples and sequencing
Six eastern gorillas were sequenced as part of this study. Five Bwindi mountain gorillas were sampled after death by the Mountain Gorilla Veterinary Project. One Mount Tshiaberimu individual was sampled under anaesthesia. Convention on the Trade in Endangered Species of Wild Fauna and Flora (CITES) permits were obtained for all samples. Sequencing was performed on the Illumina HiSeq X platform. Detailed information on all samples is provided in the Supplementary Materials (Extended Data Table 1).

### Data processing
We integrated the newly sequenced samples alongside previously published, high-coverage genomic data[7,8]. Raw sequencing reads were mapped to the human hg19 reference genome, as described previously[52]. Given that the hg19 reference does not belong to any of the gorilla subspecies, equal mapping bias will be exerted across all gorillas in our dataset. This would not have been the case if the gorilla reference genome was used instead, as it might have introduced bias in amounts of allele sharing, as observed previously for chimpanzees and bonobos[52]. The final dataset derives from 49 gorillas of known subspecies: 12 mountain (*Gorilla beringei beringei*), 9 eastern lowland *(Gorilla beringei graueri)*, 1 Cross River *(Gorilla gorilla diehli)* and 27 western lowland *(Gorilla gorilla gorilla)* gorillas.

Processing of data to obtain genotypes followed procedures described in ref. 22. We used bcftools to retain genotypes with a coverage larger than fivefold and lower than 101-fold, a mapping quality over 20, a proportion of MQ0 reads <10% and an allele balance >0.1 at heterozygous positions; bedtools and jvarkit[53] to filter the data by known repeats (RepeatMasker) and mappability (35 $k$-mer). Following a previous study[22], we used the rhesus macaque reference genome (Mmul10) to infer ancestral allele states at each site and generate an ancestral binary genome, as implemented in the freezing-archer repository (https://github.com/bvernot/freezing-archer). Only positions with genotype information in all individuals after filtering were used for calculating summary statistics for the demographic model and the $S^*$ analysis. For hmmix, missing data were allowed, genotypes were filtered for known repeats and mappability and then an individual-based filtering was applied for sequencing coverage (depth 6–100), mapping quality (20) and retained only biallelic single nucleotide variants.

### Demographic modelling
**Null demographic model.** To infer a reliable null demographic model for the four extant gorilla subspecies, we performed ABC modelling using the R package abc[54] with neural networks, following a previously described strategy[22]. Previous demographic models did not include all of the four extant gorilla subspecies[8,13]. We first attempted a merging of these models (Supplementary Table 6) but in simulations this proved a poor fit to the empirical data in terms of the distributions of segregating sites, one of the main determinants of $S^*$ (Supplementary Fig. 3).

We used ms[55] to simulate data and aimed to generate 35,700 coalescent simulation replicates, of which 35,543 were successful, whereby per iteration we generated 2,500 windows of length 40 kb, randomly sampling from wide uniform priors informed by refs. 8,13,35 (Supplementary Table 2). We sampled local mutation rates from a normal distribution with mean of $1.235 \times 10^{-8}$ (mutation rate per generation), recombination rate from a negative binomial distribution with mean of $9.40 \times 10^{-9}$ and gamma of 0.5 and assumed a generation time of 19 years (ref. 35). We scaled the mean mutation rate to 1.976 ($1.235 \times 10^{-8} \times$ window size of 40 kb $\times$ 4 $\times N_e$ of 1,000) with a scaled standard deviation of

0.460408 (1.976 × 0.233). We also scaled mean recombination rate to 1.504 ($9.40 \times 10^{-9} \times 4 \times N_e$ of 1,000 × window size of 40 kb). Per window and per population, we calculated the following summary statistics: mean and standard deviations of heterozygosity, nucleotide diversity (pi) and Tajima's $D$, as well as the number of population-wise fixed and segregating sites, the number of fixed sites per individual and pairwise $F_{ST}$ (Supplementary Table 5). These measures constitute the input summary statistics for all ABC analyses performed in this section. Given that only one diploid sample is available for *G. gorilla diehli*, we did not use standard deviations of heterozygosity, nucleotide diversity and fixed sites per individual, as well as mean nucleotide diversity for this population.

We calculated the equivalent summary statistics normalized by data coverage for the empirical data, which had been prefiltered by repeats, mappability and sufficiently informative windows (>50% of sites with confident genotype calls in all individuals). We also filtered by sites fixed across all gorillas relative to the human reference genome. We accepted parameter values from the prior distribution if they generated summary statistics close to those of the empirical data. This was assessed using a tolerance of 0.005, logit transformation of all parameters and 100 neural networks in the ABC analysis.

**Alternative demographic models.** We performed parameter inference for two further demographic models, in which we allowed gene flow from a 'ghost' lineage into the common ancestor of (B) eastern gorillas (*G. beringei beringei* and *G. beringei graueri*) (Supplementary Table 2) and (C) western gorillas (*G. gorilla gorilla* and *G. gorilla diehli)* (Supplementary Table 2). For each alternate demographic model, as above, we performed ABC analysis using 35,700 simulation replicates, whereby per iteration we generated 2,500 windows of length 40 kb. We fixed parameters with narrow CrIs from model A, to reduce the complexity of these models. To assess the impact of fixing well-inferred parameters from the null model on subsequent ghost parameter inference and explore the ghost parameter space more fully we undertook a revised modelling approach (Supplementary Section 3.4). In these revised ghost models, we performed parameter inference sampling all parameters from priors, for ghost gene flow into the common ancestor of (D) eastern gorillas and (E) western gorillas (Supplementary Table 2). We observed a strong correlation between the estimated parameters of the original and the revised ghost models, albeit with wider posterior distributions for the revised models due to increased complexity and larger parameter space (Supplementary Section 3.4).

To compare the three main demographic models—(A) null demography, (B) ghost gene flow into the eastern common ancestor and (C) ghost gene flow into the western common ancestor—we simulated 10,000 replicates of 250 windows of 40 kb length, fixing the parameters as the weighted median posteriors for each model. To achieve an equal simulated timeframe (number of generations) in all models under comparison, we added a non-interacting ghost population to the null demography, with a divergence time between ghost and extant gorillas equal to that inferred under Model B above. To determine if the models could be differentiated from each other we performed cross-validation with the function cv4postpr (nval = 1,000, tol = 0.05, method = "neural-net"). We calculated the posterior probabilities of each demographic model using the function postpr (tol = 0.05, method = "neuralnet"). The resulting confusion matrix is shown in Supplementary Table 3. We also performed cross-validation and model comparison for the five demographic models: (A) null demography, (B) ghost gene flow into the eastern common ancestor, (C) ghost gene flow into the western common ancestor, (D) revised model of ghost gene flow into the eastern common ancestor and (E) revised model of ghost gene flow into the western common ancestor, where we still observed model B having the highest support (Supplementary Table 4 and Supplementary Section 3.4).

## Detecting introgressed fragments

Following refs. [20,22,29] we calculated the $S^*$ statistic using a customized version of the package freezing-archer, accommodating non-human samples. We calculated the $S^*$ statistic genome-wide in 40 kb windows, sliding every 30 kb, using the following test (i = ingroup) and reference (o = outgroup) populations: (1) GBG (*G. beringei graueri*-i and *G. gorilla gorilla*-o) and (2) GBB (*G. beringei beringei*-i and *G. gorilla gorilla*-o). For the $S^*$ analysis, 15,181,832 variants were included.

Identifying outliers for the $S^*$ statistic requires a distribution of scores for local mutation densities (represented by numbers of segregating sites in the dataset) under a demographic scenario without introgression, as the null model. We used the weighted median posteriors for each parameter value from the above ABC analysis to generate simulated data, specifying the number of segregating sites in a stepwise manner (from 15 to 800 in steps of 5). For each stepwise segregating site (158 in total), we simulated 20,000 windows of length 40 kb, to which we applied the $S^*$ statistic for each of the scenarios (GBG and GBB). From this we obtained generalized additive models (GAMs) per scenario for three CIs (95%, 99% and 99.5%) using the R package mgcv, following the procedures described in detail in refs. [22,29]. From these GAMs, we predicted the expected $S^*$ distributions under the null model without archaic introgression. Applying the GAMs to the empirical data we inferred whether any windows lay outside the expectation per scenario and per confidence interval, assessing CIs of 95%, 99% and 99.5%. As such, the threshold of significance is defined as the 95%, 99% or 99.5% CI from the standard deviation for expected $S^*$ scores, given the mutation density[22,29].

To assess the performance of the $S^*$ statistic under our null model and its robustness to model misspecifications, we performed validation analyses following ref. [31], using msprime[56,57] simulations with explicit tracking of the introgressed fragments. Briefly, we simulated expected distributions of $S^*$ scores for the null model (model A) and for a model where the effective population sizes before 40 ka were set to the upper end of the 95% CrI ('worst' null model, in terms of highest expected amount of incomplete lineage sorting). We then simulated datasets of ten outgroup individuals (western lowland gorillas) and a single ingroup individual (eastern lowland or mountain gorillas) for model B and a model where the effective population sizes before 40 ka were set to the upper end of the 95% CI ('worst' model B). We then obtained putatively introgressed fragments using the expected scores from either model A or the 'worst' null model (Supplementary Table 7). For each model, we performed 100 replicates and calculated the average precision and recall at different thresholds. For the $S^*$ approach, we used the quantiles of the $S^*$ statistic as thresholds, which range from 0 to 0.999.

In an independent approach to the $S^*$ statistic, we applied hmmix[30]. We obtained the input files for this method: weight files, local mutation rates and individual observations files using scripts provided with the repository for hmmix (https://github.com/LauritsSkov/Introgression-detection, as of 2 August 2018), as well as bcftools, bedtools, jvarkit and custom R scripts. The macaque allele (RheMac10 assembly) was used for polarization of alleles. We then applied the method to the eastern gorillas using the following prior parameters: starting_probabilities = [0.98, 0.02], transitions = [[0.9995, 0.0005], [0.012, 0.988]], emissions = [0.05, 0.5]. We confirmed that using different parameters did not affect the results. We used a recombination rate of $9.40 \times 10^{-9}$ per site per generation and 19 years generation time with the median fragment length to estimate introgression time. Decoding, that is assigning internal and external states to specific genomic regions, was done with the script provided with the repository. Putative external fragments were filtered for posterior probabilities of 0.9 (lenient) or 0.95 (strict) and required to contain at least five private positions. We also conducted a performance analysis of hmmix on introgressed fragments in simulations of either model B or the 'worst' model B, with results similar to those for $S^*$ (Supplementary Table 8).

For performance testing the hmmix approach, we used the posterior probabilities estimated by hmmix as thresholds, which range from 0 to 0.9999.

We note that only hmmix could be used to infer archaic introgressed fragments on the X chromosome, due to the lack of a gorilla demographic model for the sex chromosomes.

## Exploring introgressed regions

To obtain a consensus set of putative introgressed regions, we overlapped the autosomal outlier regions inferred under the two methods within each eastern gorilla. For this overlap, we calculated the percentage of overlapping base pairs, considering in turn each $S^*$ confidence interval (95%, 99% and 99.5%) and with and without a 40 kb length cutoff for hmmix regions identified under the strict threshold. Imposing a 40 kb length cutoff retains 76.7% of the total strict hmmix regions (Supplementary Tables 9 and 11). We consider the intersect of the $S^*$ 99% outliers with the strict hmmix autosomal outliers, as our putative introgressed regions of high confidence. To determine whether the overlap obtained differed from random expectation we generated intersections of random regions, of equivalent distribution to the empirical data, for 100 iterations.

As a proxy for gene density we calculated the proportion of protein-coding base pairs within these regions of high confidence. As above, we compared this to the proportion of protein-coding base pairs within 100 iterations of random genomic regions, of equal length distribution as the putative introgressed regions within each eastern gorilla. We calculated pairwise nucleotide differences between individuals in the putative introgressed regions and in random genomic regions of equal length distribution and sufficient callable sites. This was conducted for three comparisons: (1) among eastern gorillas, (2) among western gorillas and (3) between eastern and western gorillas.

To assess mutational tolerance, we used GERP, SIFT, PolyPhen-2 and LINSIGHT scores[58–61]. We calculated the proportion of high impact sites for GERP, SIFT and PolyPhen-2 scores and the mean LINSIGHT score within putative introgressed regions and random regions of equal length distribution and sufficient callable sites. To explore the impact of introgression on regulatory elements, we calculated the proportion of regulatory base pairs using gorilla-defined regulatory element annotations[33], within putative introgressed and random regions of equivalent length and callability. This was assessed globally and per regulatory element type, considering poised, strong and weak, enhancers and promoters.

We further explored our putative introgressed regions of high confidence using PCA (Supplementary Fig. 13). This was generated using the biallelic sites in our putative introgressed regions. For comparison, we also generated PCAs of one random set of random regions, with equal length distribution of random regions as the putative introgressed regions per eastern gorilla. The PCAs in Fig. 1 were generated using biallelic SNPs of random genomic regions of equivalent length distribution to the putative introgressed regions of GBB Bwiruka. This sample of random genomic regions is representative of the whole genome. All PCAs were generated with the R package adegenet[62]. We generated phylogenetic trees of our putative introgressed regions and one random replicate (Supplementary Fig. 14), using the 'K80' model of nucleotide substitution, using the adegenet package[63]. Haplotype networks were drawn using pegas[64].

We localized introgression deserts by screening 1 Mb non-overlapping windows (bins) spanning the genome. We filtered out bins overlapping centromeres and those at the end of each chromosome which were <1 Mb in size. Per bin we calculated the frequency of putative introgressed regions falling within the bin, for each eastern gorilla. We also calculated data coverage of the bins and filtered by mean callable proportion >0.5. Deserts hence constitute bins where no eastern gorilla carried a putative introgressed region and which had a reasonable number of callable sites.

Plots were created with ggplot2 (ref. 65), circlize[66] and pheatmap (https://github.com/raivokolde/pheatmap). Genomic ranges were analysed with the GenomicRanges package[67].

## Adaptive introgression

To explore signatures of adaptive introgression within eastern gorillas, we applied the genome-wide scan VolcanoFinder[34]. To do so, we polarized the data to two outgroups. First, we polarized the human reference allele using the rhesus macaque allele and subsequently polarized the gorilla genotypes by this polarized allele representing the ancestral state. To obtain the allele frequency input files per chromosome, we then filtered our data to only eastern gorilla genotypes at biallelic sites and also filtered out sites with multiple ancestral alleles (where polarization would be uncertain) and sites of reference homozygotes. The second input file required is an empirical unnormalized site frequency spectrum (SFS), which we generated by obtaining the unfolded SFS, normalizing so all site categories sum to 1 and then filtering out the first category (the 0 entry). We called VolcanoFinder specifying '-big 1000, $D = -1$, $P = 1$, Model = 1'. For computational efficiency, we performed the VolcanoFinder scan in blocks, whereby each chromosome was split into blocks of approximately equal numbers of base pairs. We placed a test site every 1,000 bp (-big 1000). We set $D$ to $-1$, so VolcanoFinder iteratively tested a grid of values for genetic distance internally and selected the value that maximizes the likelihood ratio[34]. We set $P$ to 1 as our input data were polarized. We used Model = 1, following procedures applied to human data[34], as well as non-human species[68,69].

We took the 95% outliers of composite likelihood ratio scores calculated from VolcanoFinder and intersected these regions with our putative introgressed regions (identified above), to obtain putative adaptive introgressed targets. To explore potential functional consequences, we assessed which genes and which mutations fall within the putative adaptive introgressed targets, using the Variant Effect Predictor annotation (v.83)[70].

## Reporting summary

Further information on research design is available in the Nature Portfolio Reporting Summary linked to this article.

# Data availability

The six newly sequenced eastern gorilla samples are publicly available in the European Nucleotide Archive (ENA) under the project number: PRJEB12821. ENA accession numbers for all samples used in this study are given in Extended Data Table 1. The human reference genome (hg19) and the rhesus macaque reference genome (Mmul10/rheMac10) were downloaded from https://hgdownload.soe.ucsc.edu/goldenPath/. Precalculated GERP scores for hg19 were accessed from http://mendel.stanford.edu/SidowLab/downloads/gerp/ and LINSIGHT scores for hg19 from https://rdrr.io/github/rcastelo/GenomicScores/src/inst/scripts/make-data_linsight.UCSC.hg19.R.

# Code availability

Scripts used for data analysis are available on Github under https://github.com/h-pawar/gor_ghost_introg.

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

## Acknowledgements

We thank D. Setter for valuable guidance in applying VolcanoFinder. We thank the Uganda Wildlife Authority for the Gorilla monitoring and research permission. We are grateful to the Life Science Compute Cluster of the University of Vienna. This project has been funded by the Vienna Science and Technology Fund (WWTF) (grant no. 10.47379/VRG20001) to M.K. and the European Research Council under the European Union's Horizon 2020 research and innovation programme (grant no. 864203), PID2021-126004NB-100 (MINECO/FEDER, UE), Secretaria d'Universitats i Recerca and CERCA Program del Departament d'Economia i Coneixement de la Generalitat de Catalunya (GRC 2021 SGR 00177) to T.M.-B. H.P. was supported by a Formació de Personal Investigador fellowship from Generalitat de Catalunya (FI_B100131). M.A.-E. was supported by a Formación de Personal Investigador PRE2018-083966 from Ministerio de Ciencia, Universidades e Investigación. C.T.-S., Y.X. and J.P.-M. were funded by Wellcome grant no. 098051. K.G. was supported by Swedish Research Council grant no. 2020-03398. J.L.K. received the María de Maeztu Mobility Fellowship. O.D. was supported by a John Templeton Foundation grant no. ID 62178. A.M.A. received funding from UCL's Wellcome Trust ISSF3 award no. 204841/Z/16/Z. Q.A. is supported by strategic funding from Monash University (STG-000114).

## Author contributions

T.M.-B. and M.K. conceived and conceptualized the study. H.P. performed demographic modelling. Q.A. performed experiments. H.P., A.R., M.d.M., T.v.d.V., I.L., M.H., R.B., O.D., S.H., P.E.-C., J.P.-M. and M.K. analysed data. X.H., O.L. and D.J. provided software. K.G., A.M.A., C.T.-S., Y.X., T.M.-B. and M.K. provided supervision. B.S., M.C., C.T.-S. and Y.X. acquired samples and their documentation. M.A.-E. visualized results. S.C.-E., J.L.K. and O.E.C. provided comments. H.P., T.M.-B. and M.K. wrote the paper with input from all authors.

## Competing interests

The authors declare no competing interests.

## Additional information

**Extended data** is available for this paper at https://doi.org/10.1038/s41559-023-02145-2.

**Correspondence and requests for materials** should be addressed to Tomas Marques-Bonet or Martin Kuhlwilm.

[1]Institute of Evolutionary Biology (UPF-CSIC), PRBB, Barcelona, Spain. [2]Department of Evolutionary Anthropology, University of Vienna, Vienna, Austria. [3]Human Evolution and Archaeological Sciences (HEAS), University of Vienna, Wien, Austria. [4]Department of Bioinformatics and Genetics, Scilifelab, Swedish Museum of Natural History, Stockholm, Sweden. [5]Centre for Palaeogenetics, Stockholm, Sweden. [6]Institute of Cancer and Genomic Sciences, University of Birmingham, Dubai, United Arab Emirates. [7]Integrative Genomics Lab, CIC bioGUNE—Centro de Investigación Cooperativa en Biociencias, Parque Científico Tecnológico de Bizkaia building 801A, Derio, Spain. [8]Wellcome Sanger Institute, Hinxton, UK. [9]Monash University Malaysia Genomics Facility, School of Science, Monash University Malaysia, Selangor Darul Ehsan, Malaysia. [10]Department of Ecology and Evolutionary Biology, University of California, Santa Cruz, CA, USA. [11]UCL Genetics Institute, Department of Genetics, Evolution and Environment, University College London, London, UK. [12]Animal Ecology, Department of Ecology and Genetics, Uppsala University, Uppsala, Sweden. [13]Institute of Ecology and Evolution, School of Biological Sciences, University of Edinburgh, Edinburgh, UK. [14]Science for Life Laboratory, Uppsala, Sweden. [15]Gorilla Doctors, Kampala, Uganda. [16]Gorilla Doctors, Karen C. Drayer Wildlife Health Center, One Health Institute, University of California Davis, School of Veterinary Medicine, Davis, CA, USA. [17]Catalan Institution of Research and Advanced Studies (ICREA), Passeig de Lluís Companys, Barcelona, Spain. [18]CNAG-CRG, Centre for Genomic Regulation (CRG), Barcelona Institute of Science and Technology (BIST), Barcelona, Spain. [19]Institut Català de Paleontologia Miquel Crusafont, Universitat Autònoma de Barcelona, Edifici ICTA-ICP, Barcelona, Spain. [20]These authors contributed equally: Tomas Marques-Bonet, Martin Kuhlwilm. ✉e-mail: tomas.marques@upf.edu; martin.kuhlwilm@univie.ac.at

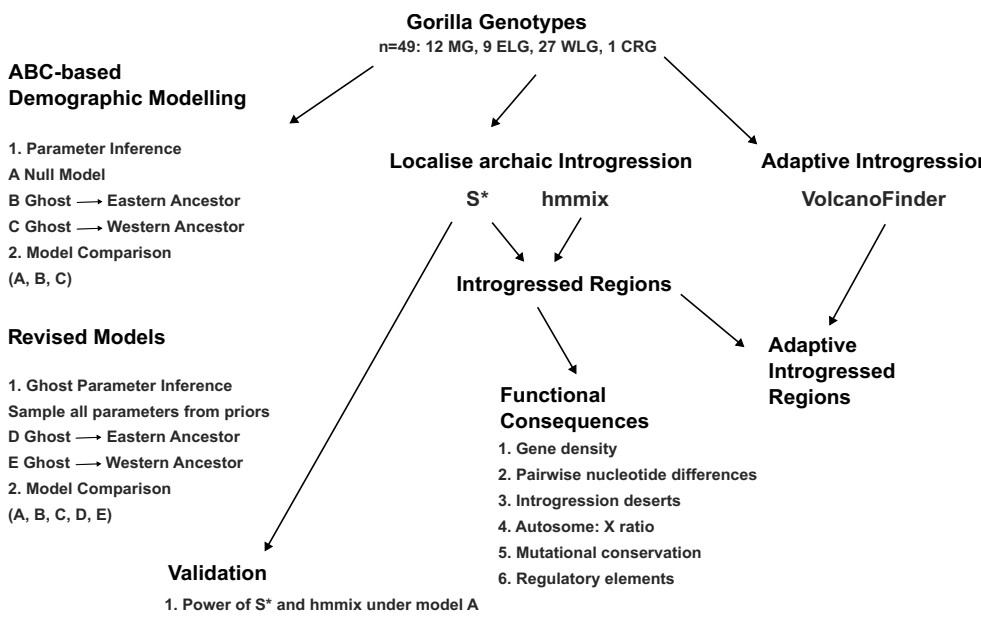

**Extended Data Fig. 1 | Workflow of the main analyses.**

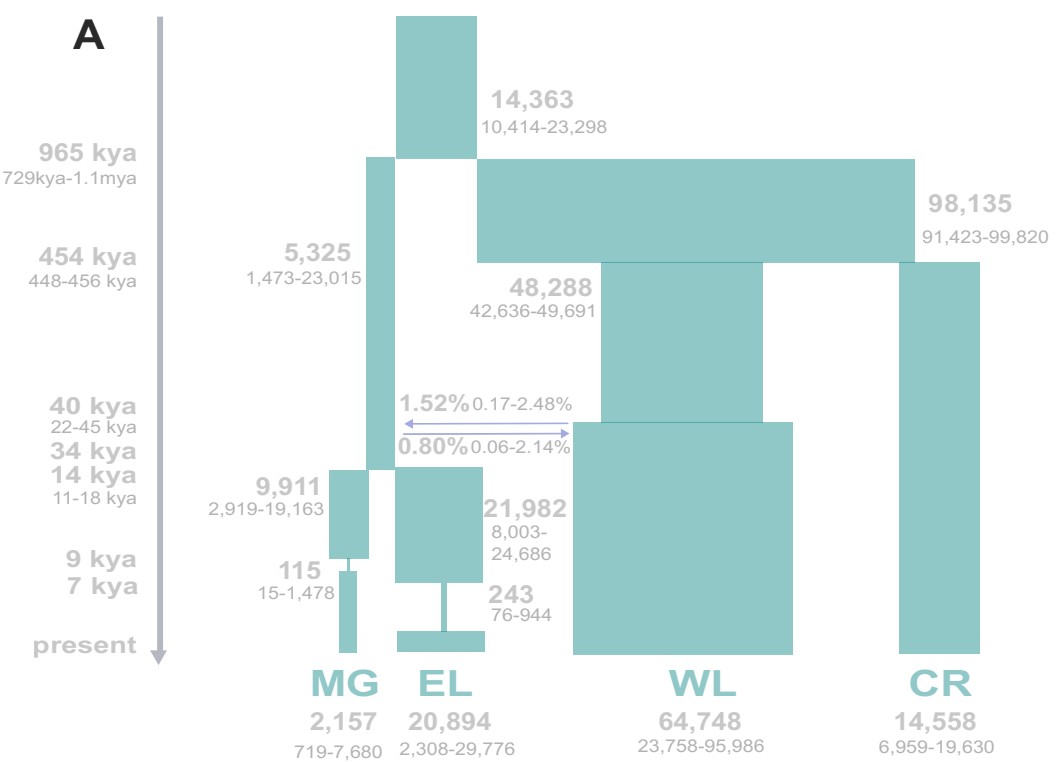

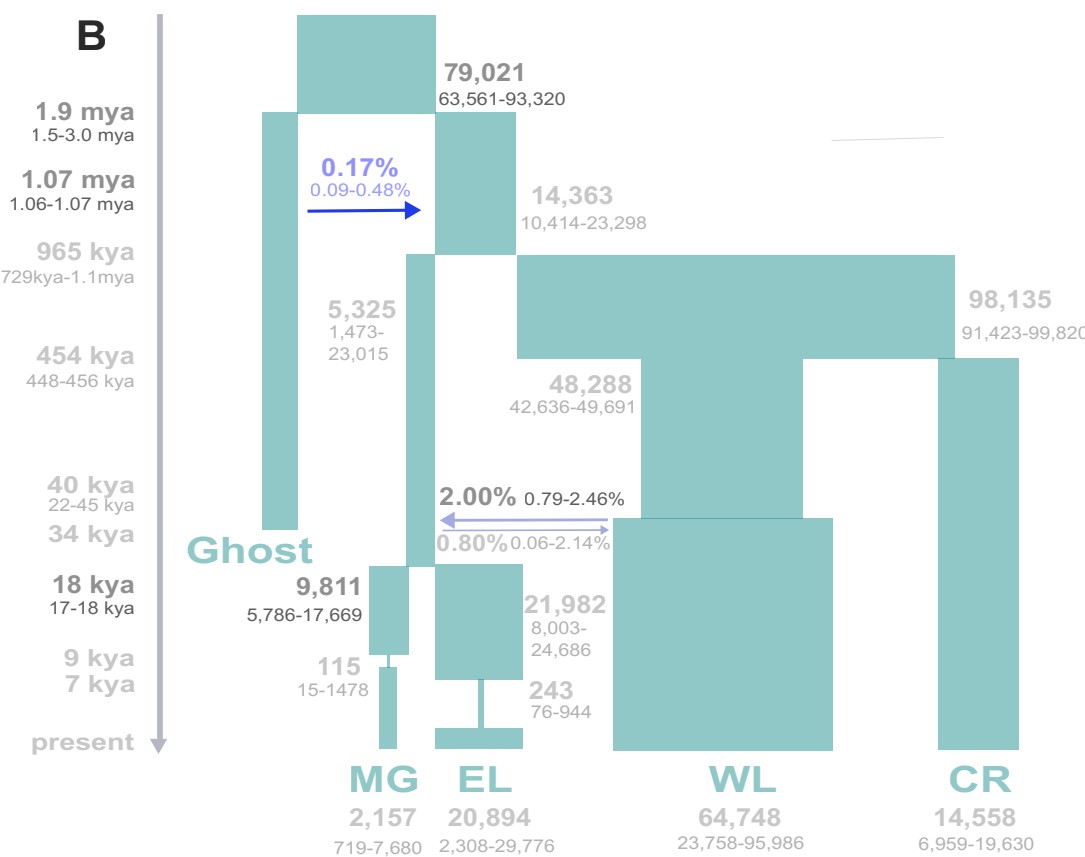

**Extended Data Fig. 2 | Demographic models A and C. A** Null model of gorilla population history (only extant populations). 95% credible intervals are shown for all parameters inferred. **B** Alternate model allowing the possibility of ghost introgression into the common ancestor of western gorillas, resulted in a model of ancestral population structure being inferred. We note under a model of ghost gene flow to the western common ancestor, the posteriors indicate a small contribution to the common ancestor of all gorillas (consistent with ancestral substructure), rather than a defined pulse to the western common ancestor. In darker colours are the parameters inferred under this alternate model with their 95% credible intervals.

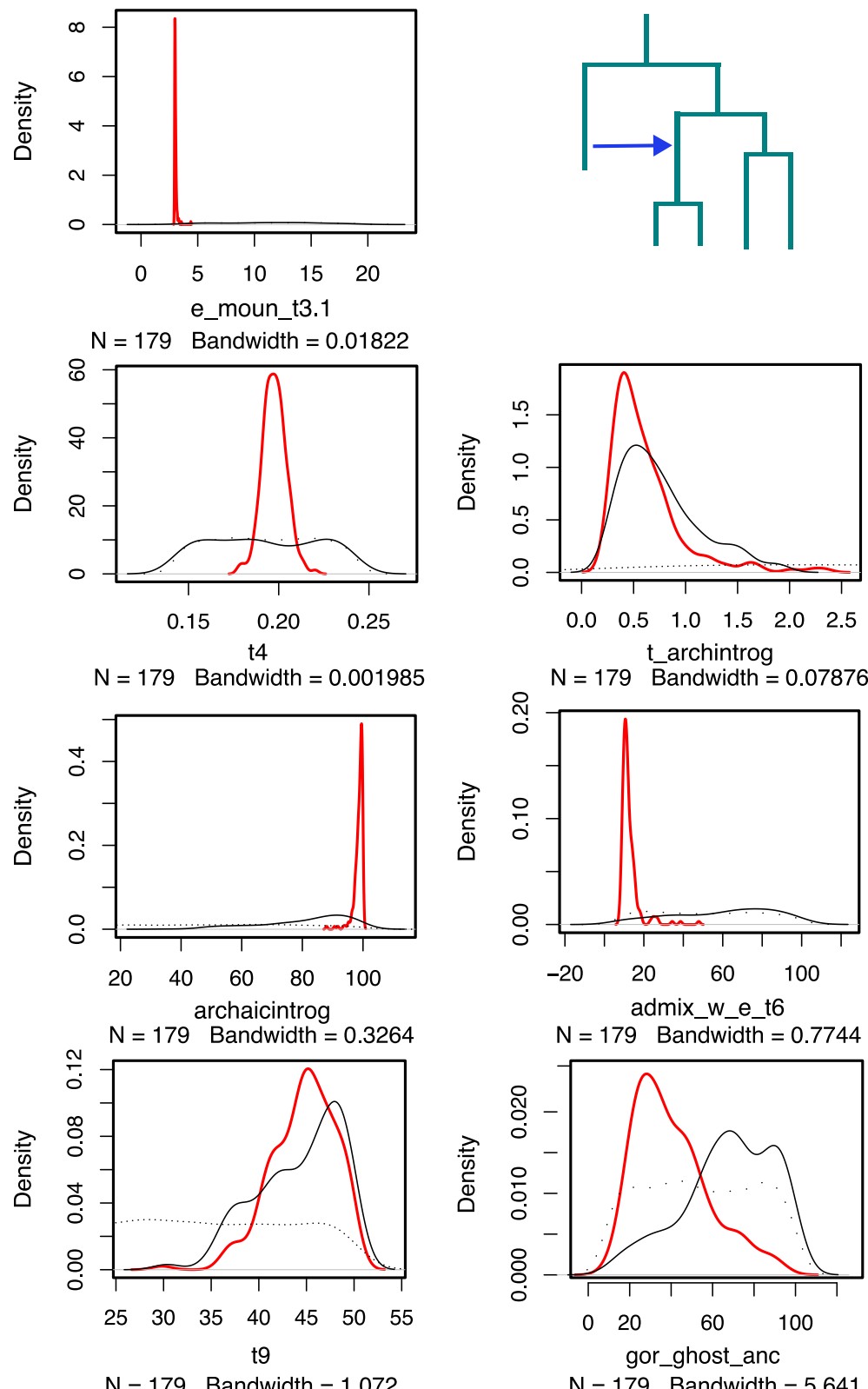

**Extended Data Fig. 3 | Prior and posterior distributions for model B.** Parameter distributions for all parameters inferred under the ABC model allowing gene flow from a ghost lineage into the common ancestor of eastern gorillas. Red indicates the posterior distribution inferred with neural networks. Black indicates the posterior distribution inferred under a rejection method. The dotted grey line indicates the prior distribution.

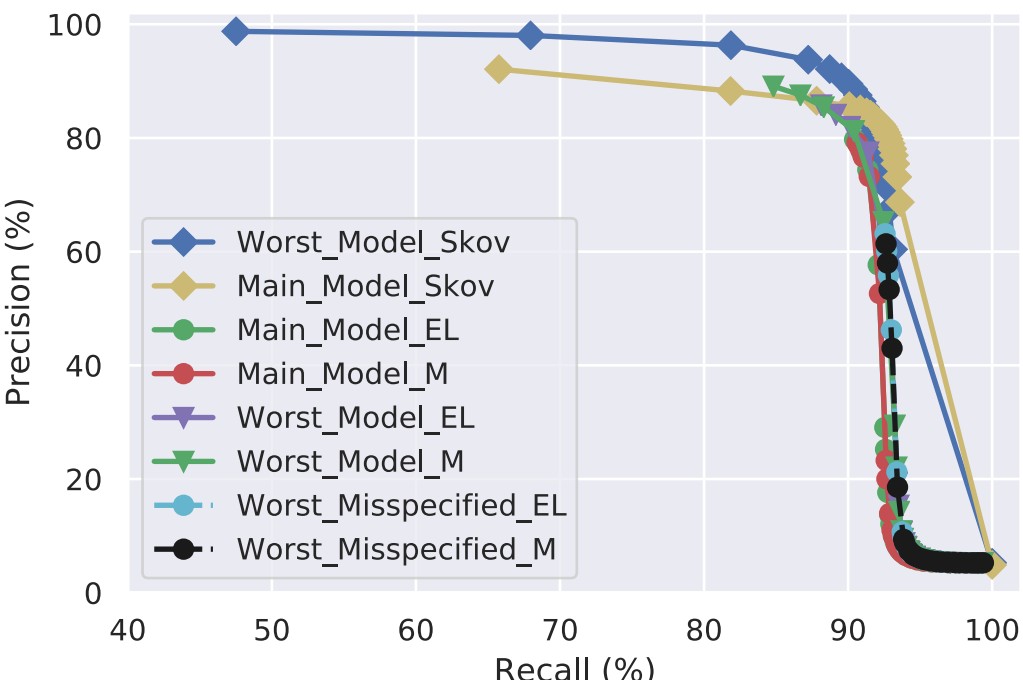

**Extended Data Fig. 4 | Performance of *S*\* and hmmix.** Precision-recall curves for the *S*\* statistic as implemented in sstar[31] and for hmmix. Main model refers to a model taking the weighted median posteriors from the ABC-based null demography presented herein (Extended Data Fig. 2A). Worst model refers to a model taking the maximum value of the 95% credible interval for all ancestral Ne parameters from the ABC-based null demography. For the *S*\* statistic we consider the target population as alternately eastern lowland or mountain gorillas, eg Main Model EL. Worst mis-specified is where we generate simulated data under the worst model but run the *S*\* analysis using the 'quantile' or outlier values inferred under the main model. Skov=hmmix method, EL=eastern lowland gorillas, M=mountain gorillas.

**Extended Data Table 1 | Information for gorillas analysed herein. 49 samples: 12 *Gorilla beringei beringei* (mountain gorillas), 9 *Gorilla beringei graueri* (eastern lowland gorillas), 1 *Gorilla gorilla diehli* (Cross River gorillas), 27 *Gorilla gorilla gorilla* (western lowland gorillas)**

| Subspecies | Individual | SRA ID | Sex | Country or origin | Project |
|---|---|---|---|---|---|
| Mountain gorilla | Bwiruka | ERR2300765 | F | Uganda (Bwindi) | this paper |
| Mountain gorilla | Imfura | ERS168207 | M | Rwanda (Virunga) | Xue *et al.* 2015 |
| Mountain gorilla | Kaboko | ERS168410 | M | DRC (Virunga) | Xue *et al.* 2015 |
| Mountain gorilla | Kahungye | ERR2300762 | F | Uganda (Bwindi) | this paper |
| Mountain gorilla | Katungi | ERR2300763 | F | Uganda (Bwindi) | this paper |
| Mountain gorilla | Maisha | ERS525616 | F | DRC (Virunga) | Xue *et al.* 2015 |
| Mountain gorilla | Nyamunwa | ERR2300764 | F | Uganda (Bwindi) | this paper |
| Mountain gorilla | Semehe | ERR2300766 | F | Uganda (Bwindi) | this paper |
| Mountain gorilla | Tuck | ERS168204 | F | Rwanda (Virunga) | Xue *et al.* 2015 |
| Mountain gorilla | Turimaso | ERS525618 | F | Rwanda (Virunga) | Xue *et al.* 2015 |
| Mountain gorilla | Umurimo | ERS525617 | F | Rwanda (Virunga) | Xue *et al.* 2015 |
| Mountain gorilla | Zirikana | ERS168174 | M | Rwanda (Virunga) | Xue *et al.* 2015 |
| Eastern lowland gorilla | 9732_Mkubwa | SRS396825 | M | DRC (Tulakwa) | Prado-Martinez *et al.* 2013 |
| Eastern lowland gorilla | A929_Kaisi | SRS396605 | M | DRC (Walikale) | Prado-Martinez *et al.* 2013 |
| Eastern lowland gorilla | A967_Victoria | SRS396876 | F | DRC | Prado-Martinez *et al.* 2013 |
| Eastern lowland gorilla | Dunia | ERS525621 | F | DRC (Walikale) | Xue *et al.* 2015 |
| Eastern lowland gorilla | Itebero | ERS168205 | F | DRC (Kahuzi-Biega) | Xue *et al.* 2015 |
| Eastern lowland gorilla | Mukokya | ERR2300767 | M | DRC (Mount Tshiaberimu) | this paper |
| Eastern lowland gorilla | Ntabwoba | ERS168206 | M | DRC (Walikale) | Xue *et al.* 2015 |
| Eastern lowland gorilla | Pinga | ERS525620 | F | DRC | Xue *et al.* 2015 |
| Eastern lowland gorilla | Tumani | ERS525619 | F | DRC (Walikale) | Xue *et al.* 2015 |
| Cross river gorilla | B646_Nyango | SRS396855 | F | West Africa | Prado-Martinez *et al.* 2013 |
| Western lowland gorilla | 9749_Kowali | SRS396819 | F | unknown | Prado-Martinez *et al.* 2013 |
| Western lowland gorilla | 9750_Azizi | SRS396821 | M | Cameroon | Prado-Martinez *et al.* 2013 |
| Western lowland gorilla | 9751_Bulera | SRS396820 | F | Cameroon | Prado-Martinez *et al.* 2013 |
| Western lowland gorilla | 9752_Suzie | SRS394796 | F | unknown | Prado-Martinez *et al.* 2013 |
| Western lowland gorilla | 9753_Kokomo | SRS396849 | F | unknown | Prado-Martinez *et al.* 2013 |
| Western lowland gorilla | A930_Sandra | SRS396606 | F | Cameroon | Prado-Martinez *et al.* 2013 |
| Western lowland gorilla | A931_Banjo | SRS396826 | M | Cameroon | Prado-Martinez *et al.* 2013 |
| Western lowland gorilla | A932_Mimi | SRS396827 | F | Cameroon | Prado-Martinez *et al.* 2013 |
| Western lowland gorilla | A933_Dian | SRS396828 | F | Cameroon | Prado-Martinez *et al.* 2013 |
| Western lowland gorilla | A934_Delphi | SRS396829 | F | Congo | Prado-Martinez *et al.* 2013 |
| Western lowland gorilla | A936_Coco | SRS396830 | F | Equatorial Guinea | Prado-Martinez *et al.* 2013 |
| Western lowland gorilla | A937_Kolo | SRS396831 | F | Cameroon | Prado-Martinez *et al.* 2013 |
| Western lowland gorilla | A962_Amani | SRS396847 | F | unknown | Prado-Martinez *et al.* 2013 |
| Western lowland gorilla | B642_Akiba_Beri | SRS396852 | F | Cameroon | Prado-Martinez *et al.* 2013 |
| Western lowland gorilla | B643_Choomba | SRS396853 | F | West Africa | Prado-Martinez *et al.* 2013 |
| Western lowland gorilla | B644_Paki | SRS396854 | F | West Africa | Prado-Martinez *et al.* 2013 |
| Western lowland gorilla | B647_Anthal | SRS396856 | F | West Africa | Prado-Martinez *et al.* 2013 |
| Western lowland gorilla | B650_Katie | SRS396857 | F | West Africa | Prado-Martinez *et al.* 2013 |
| Western lowland gorilla | KB3782_Vila | SRS396870 | F | Congo | Prado-Martinez *et al.* 2013 |
| Western lowland gorilla | KB3784_Dolly | SRS396873 | F | Congo | Prado-Martinez *et al.* 2013 |
| Western lowland gorilla | KB4986_Katie | SRS396874 | F | unknown | Prado-Martinez *et al.* 2013 |
| Western lowland gorilla | KB5792_Carolyn | SRS396868 | F | Congo | Prado-Martinez *et al.* 2013 |
| Western lowland gorilla | KB5852_Helen | SRS396871 | F | Cameroon | Prado-Martinez *et al.* 2013 |
| Western lowland gorilla | KB6039_Oko | SRS396872 | F | unknown | Prado-Martinez *et al.* 2013 |
| Western lowland gorilla | KB7973_Porta | SRS396869 | F | unknown | Prado-Martinez *et al.* 2013 |
| Western lowland gorilla | X00108_Abe | SRS396850 | M | unknown | Prado-Martinez *et al.* 2013 |
| Western lowland gorilla | X00109_Tzambo | SRS396851 | M | unknown | Prado-Martinez *et al.* 2013 |

SRA ID = Short Read Archive identifier; F = Female; M = Male; DRC = Democratic Republic of the Congo

**Extended Data Table 2 | Regions and genes with signatures of putative adaptive introgression**

| Eastern lowland gorillas | | | |
|---|---|---|---|
| Region | Genes | Minimum LR | Maximum LR |
| 12:11051000-11122000 | *TAS2R14, PRH2, TAS2R13* | 24.43 | 246.15 |
| 5:9061000-9111000, 5:9315000-9503000 | *SEMA5A* | 26.75 | 27.97 |
| 1:191807000-191925000 | RP11-541F9.2 | 25.92 | 35.06 |
| 3:124338000-124414000 | *KALRN* | 24.07 | 24.07 |
| 10:113899000-113999000 | *GPAM* | 24.99 | 30.16 |
| 7:26805000-27214000 | *SKAP2, HOXA1-10* | 24.65 | 36.19 |

| Mountain gorillas | | | |
|---|---|---|---|
| 12:11075000-11121000 | *TAS2R14, PRH2, TAS2R13* | 24.43277 | 246.15161 |
| 5:9038000-9232000, 5:9249000-9414000, 5:9436000-9503000 | *SEMA5A* | 26.75713 | 27.97717 |
| 4:69349546-69349547 | *TMPRSS11E* | 24.0434 | 24.0434 |
| 3:124318000-124414000 | *KALRN* | 24.07903 | 24.07903 |

LR = Likelihood Ratio

# Reporting Summary

## Statistics

For all statistical analyses, confirm that the following items are present in the figure legend, table legend, main text, or Methods section.

| n/a | Confirmed | |
|---|---|---|
| ☐ | ☒ | The exact sample size (*n*) for each experimental group/condition, given as a discrete number and unit of measurement |
| ☒ | ☐ | A statement on whether measurements were taken from distinct samples or whether the same sample was measured repeatedly |
| ☒ | ☐ | The statistical test(s) used AND whether they are one- or two-sided <br> *Only common tests should be described solely by name; describe more complex techniques in the Methods section.* |
| ☒ | ☐ | A description of all covariates tested |
| ☐ | ☒ | A description of any assumptions or corrections, such as tests of normality and adjustment for multiple comparisons |
| ☐ | ☒ | A full description of the statistical parameters including central tendency (e.g. means) or other basic estimates (e.g. regression coefficient) AND variation (e.g. standard deviation) or associated estimates of uncertainty (e.g. confidence intervals) |
| ☒ | ☐ | For null hypothesis testing, the test statistic (e.g. $F$, $t$, $r$) with confidence intervals, effect sizes, degrees of freedom and $P$ value noted <br> *Give P values as exact values whenever suitable.* |
| ☐ | ☒ | For Bayesian analysis, information on the choice of priors and Markov chain Monte Carlo settings |
| ☒ | ☐ | For hierarchical and complex designs, identification of the appropriate level for tests and full reporting of outcomes |
| ☒ | ☐ | Estimates of effect sizes (e.g. Cohen's *d*, Pearson's *r*), indicating how they were calculated |

*Our web collection on statistics for biologists contains articles on many of the points above.*

## Software and code

Policy information about availability of computer code

| Data collection | no software was used |
|---|---|
| Data analysis | bcftools 1.6/1.12/1.14, bedtools 2.26.0, bedops, gcc 6.3.0, hmmix, jvarkit, ms, msprime, openssl 1.0.2q, perl, python 2.7.11/2.7.14/2.7.17, R 3.2.0/3.4.0/4.0.1, samtools 1.0, sstar, tabix 0.2.6, Variant Effect Predictor 83, vcftools 0.1.15, VolcanoFinder 1.0, xz 5.2.2, zlib 1.2.8, WebGestaltR |

For manuscripts utilizing custom algorithms or software that are central to the research but not yet described in published literature, software must be made available to editors and reviewers. We strongly encourage code deposition in a community repository (e.g. GitHub). See the Nature Portfolio guidelines for submitting code & software for further information.

## Data

Policy information about availability of data

All manuscripts must include a data availability statement. This statement should provide the following information, where applicable:
- Accession codes, unique identifiers, or web links for publicly available datasets
- A description of any restrictions on data availability
- For clinical datasets or third party data, please ensure that the statement adheres to our policy

The six newly sequenced eastern gorilla samples are publicly available in the European Nucleotide Archive (ENA) under the project number: PRJEB12821. ENA accession numbers for all samples used in this study are given in Table S1. The human reference genome (hg19) and the rhesus macaque reference genome

## Human research participants

Policy information about studies involving human research participants and Sex and Gender in Research.

| | |
|---|---|
| Reporting on sex and gender | Not applicable |
| Population characteristics | Not applicable |
| Recruitment | Not applicable |
| Ethics oversight | Not applicable |

Note that full information on the approval of the study protocol must also be provided in the manuscript.

# Field-specific reporting

Please select the one below that is the best fit for your research. If you are not sure, read the appropriate sections before making your selection.

☒ Life sciences    ☐ Behavioural & social sciences    ☐ Ecological, evolutionary & environmental sciences

For a reference copy of the document with all sections, see nature.com/documents/nr-reporting-summary-flat.pdf

# Life sciences study design

All studies must disclose on these points even when the disclosure is negative.

| | |
|---|---|
| Sample size | 49 whole genomes of endangered wild species. No more samples could be obtained. |
| Data exclusions | One sample (Nkuhene) was excluded due to very low quality, with 80% of read duplicates and 2X average coverage. |
| Replication | Not applicable, as the 49 genomes are independent individuals from the wild, as representatives of their respective populations. |
| Randomization | Not applicable, as these are wild individuals, and no experimental conditions apply. |
| Blinding | Not applicable, as no experimental conditions apply. |

# Reporting for specific materials, systems and methods

We require information from authors about some types of materials, experimental systems and methods used in many studies. Here, indicate whether each material, system or method listed is relevant to your study. If you are not sure if a list item applies to your research, read the appropriate section before selecting a response.

## Materials & experimental systems

| n/a | Involved in the study |
|---|---|
| ☒ | ☐ Antibodies |
| ☒ | ☐ Eukaryotic cell lines |
| ☒ | ☐ Palaeontology and archaeology |
| ☒ | ☐ Animals and other organisms |
| ☒ | ☐ Clinical data |
| ☒ | ☐ Dual use research of concern |

## Methods

| n/a | Involved in the study |
|---|---|
| ☒ | ☐ ChIP-seq |
| ☒ | ☐ Flow cytometry |
| ☒ | ☐ MRI-based neuroimaging |

