## [Peer Review File · Nature Ecology & Evolution]

Peer Review Information

Journal: Nature Ecology & Evolution

Manuscript Title: Ghost admixture in eastern gorillas

Corresponding author name(s): Tomas Marques-Bonet, Martin Kuhlwilm

Editorial Notes:

Reviewer Comments & Decisions:

Decision Letter, initial version:

6th February 2023

Dear Professor Kuhlwilm,

Your manuscript entitled "Ghost admixture in eastern gorillas" has now been seen by three reviewers, whose comments are attached. The reviewers have raised a number of concerns which will need to be addressed before we can offer publication in Nature Ecology & Evolution. We will therefore need to see your responses to the criticisms raised and to some editorial concerns, along with a revised manuscript, before we can reach a final decision regarding publication.

In particular, we note that while the reviewers are overall positive about the manuscript, in addition to technical criticisms there are some concerns about novelty/degree of advance. To mitigate this, we recommend that you attend to reviewer 3's recommendations for additional analyses that could offer some additional novel results that go beyond the previous work with other ape species.

We therefore invite you to revise your manuscript taking into account all reviewer and editor comments. Please highlight all changes in the manuscript text file.

- * Include a "Response to reviewers" document detailing, point-by-point, how you addressed each reviewer comment. If no action was taken to address a point, you must provide a compelling argument. This response will be sent back to the reviewers along with the revised manuscript.
- * If you have not done so already please begin to revise your manuscript so that it conforms to our Article format instructions at <http://www.nature.com/natecolevol/info/final-submission>. Refer also to any guidelines provided in this letter.
- * Include a revised version of any required reporting checklist. It will be available to referees (and, potentially, statisticians) to aid in their evaluation if the manuscript goes back for peer review. A

2revised checklist is essential for re-review of the paper.

[REDACTED]

Nature Ecology & Evolution is committed to improving transparency in authorship. As part of our efforts in this direction, we are now requesting that all authors identified as 'corresponding author' on published papers create and link their Open Researcher and Contributor Identifier (ORCID) with their account on the Manuscript Tracking System (MTS), prior to acceptance. ORCID helps the scientific community achieve unambiguous attribution of all scholarly contributions. You can create and link your ORCID from the home page of the MTS by clicking on 'Modify my Springer Nature account'. For more information please visit www.springernature.com/orcid.

[REDACTED]

Reviewer expertise:

Reviewer #1: ghost lineages and admixture

Reviewer #2: human evolutionary genomics

Reviewer #3: great ape evolution and conservation genomics

Reviewers' comments:

Reviewer #1 (Remarks to the Author):

2This interesting manuscript by Pawar et al. focuses an important topic, i.e. ghost introgression, in evolutionary biology. Ghost introgression has been well documented in modern humans, in which ancient DNA from fossil remains are available, much facilitating the detection of introgression from Neanderthals and Denisovans. Using genomic data of extant species, researchers can also find the legacy of extinct species without fossil DNA. Applying the approach developed in their previous study of the Pan clade to gorilla species, the authors evaluated ghost introgression scenarios against what they called "null model" which involves gene flow only between extant gorilla species, and found the best model is ghost introgression into the common ancestor of two eastern gorilla lineages. Built on this model, they then identified the putative introgressed fragments in the genomes of eastern gorillas, particularly those fragments that are likely adaptive.

However, I have some concerns with the results and conclusions in this manuscript. It is well known that the Approximate Bayesian Computation approach requires good prior knowledge for the models to be included in analysis, because there are numerous possible ghost introgression scenarios. For example, ghost lineages may contribute genetic materials to any one of 4 subspecies rather than any common ancestor. If I were you, I would first perform some exploratory phylogenomic analyses (e.g. ABBA-BABA test or 3s or PhyloNet/Phylonetworks) to narrow down possible gene flow scenarios. In other words, the two ghost introgression models should be suggested by data, rather than simply assumed as it appeared to me. At the very least, more background information and more clarification should be provided for the model selection.

In order to reduce the complexity of the ghost introgression models in ABC, you "fixed parameters with narrow Cis from model A" (LINE 480). This is quite subjective, due to vague meaning of "narrow". Is there any sound reason for the choice? By keeping most parameters of the null model as fixed in the two ghost introgression models, you implicitly make a big assumption that ghost introgression has little influence on these parameters. At least this assumption should be justified with sufficient details. To my mind, some parameters are apparently affected by the ghost introgression events, such as the effective population size of the common ancestor of all gorillas and the divergence time between eastern and western lineages, and they should be estimated in the models.

In the manuscript, you made no reference to Table S4-S8, and some references to supplementary Tables in the manuscript seem misplaced, causing much confusion and even misunderstanding. In your Fig. S4, the ghost introgression was placed in the common ancestor of all gorillas, not in the western common ancestor as claimed in the legend or main text (see LINE 189 and LINE 191).

LINE 454: The mutation rate is per year or per generation? Please specify.

Reviewer #2 (Remarks to the Author):

Pawar et al sequence new eastern gorillas and model the demographic history of all gorilla subspecies. They then check if there is any evidence of ghost admixture into any of the common ancestors of western and eastern gorillas. They conclude that a demographic model which incorporates ghost admixture into eastern gorillas fits their summary statistics better than a model without. They

3they run two methods developed for detecting archaic introgression into modern humans. They find evidence that a deeply divergent ghost population contributed 2-3% to the genomes of eastern gorillas.

The findings are well presented and I was able to find most relevant information in the supplementary material. The authors have done a lot of work to produce these results and they provide links to the scripts in the supplementary tables which is good for rerunning analysis in the future and make the science reproducible. Good job!

The findings are interesting and I find it plausible that admixture from a ghost population into eastern gorillas took place as hybridization events are common in mammals. However I would like to see some additional analysis to ensure that these findings are robust.

Simulations

Establishing a demographic model for Gorilla species is key for this manuscript and needs some additional work.

1.1 The authors provide a detailed overview in the supplement of all their analysis and there are a lot of tabs to go through plus the naming is not intuitive. For example it is hard to remember what "e_moun_t3.1" means and for "gor_ghost_anc" the explanation is "split of all gorillas + ghost population" but the units are in individuals so it is an effective population size?

I think it could be useful to merge the tabs with priors and posteriors and have the first column be "Explanation column". I have attached a suggestion of what it could look like as "example_supp.xlsx"

1.2 I understand that the search space becomes very large so some parameters needs to be fixed but it feels like the authors might be getting at a local optimum and not a global one. For instance in tab "8.ghostw.posterior" only 7 parameters are estimated while the others are fixed. For instance the parameter "w_low_t0" is set to 64.7488. However if I look in "4.null.posterior" this parameter also has uncertainty (23.758 - 95.9855). What happens if you use this range as a prior when fitting parameters for ghost admixture?

1.3 Furthermore some parameters are always fixed from the beginning t1, t2, t3. But t4 in 2.suppl.initial.mergedsimns (5. gorilla split time (low divergence simulations) is allowed to vary. The estimate value is 965,481 years ago which is very different from previous estimates of 261,000 years ago (McManus et al. 2015) and 429,000 years ago (Sally et al. 2012). Can the authors provide a brief explanation for why some parameters are fixed while some are not?

Archiac introgression

This is the section which is most crucial to the point the authors want to make. First all models are approximations - for instance you model the effective population size of Eastern gorillas as being a constant 5,325 for almost a million years. This is of course an approximation. When one adds a ghost admixture one allows some extra flexibility in the model so scenarios with ghost admixture almost always fit the data better because there will always be some deeply divergent haplotypes present. So we have to make sure that this ghost admixture is robust to model mis-specifications.

2.1) I suggest creating msprime simulationed data using the demographic model from Fig 2 - (extended table S10). The advantage of using msprime is that you can keep track of the introgressed segments and evaluate how well you are doing in identifying them! But I would pick the values which increases the amount of deeply divergent haplotypes due to ILS and see if you can still distinguish that scenario from one with ghost admixture. This would mean increasing all effective population size estimates of ancestral populations to the maximum value of their credibility intervals e.g increasig the population size of Eastern gorillas to 23015.4 and the parameter `ne_gor_species_split` to 23298.3. Basically every population size that is before the admixture event. Let refer to this scenario as "Scenario_X" which represents a worst case scenario for identifying ghost admixture when using sstar or hmmix.

Then you could add ghost admixture of 2.5% at 38,000 years ago and lets call this scenario "Scenario_Y".

2.2) Now if you run hmmix or sstar (which ever one is easiest to run) on both scenarios using the same approach as you describe in the supplement how much "ghost admixture" does the model identify in ScenarioX vs ScenarioY?

2.3) For ScenarioY where you know which segments are actually introgressed what is the false positive, false negative rates of identifying these segments? It would also be helpful to show the length distribution of 'ghost admixture' segments for scenarioX, scenarioY and what you observe in real data like you do in Figure S2.

Additional questions

3.1 Further I was wondering why you don't train the parameters using hmmix. This should give you the coalescent time to the outgroup for 'gorilla' and 'ghost' segments. Can you provide the trained parameters? Are these consistent with the one you infer using ABC?

3.2. Why are you mapping Gorilla shotgun reads to the human reference hg19 - why not use the gorilla reference genome (gorgor6 is the latest version I believe).

3.3. You refer to S* and hmmix as being "two complementary approaches". However the main signal for both of them is SNP density of variants not found in an outgroup and the fact that these variants

5are somewhat clustered. So it is not surprising that they overlap! I would make that clear to the reader.

3.4. The authors write: "We find that 1.48-2.97% of the individual eastern gorilla genomes are inferred as external at a strict threshold for the mean probability of 0.95, with an estimated introgression time of 37-41 kya." Where does the 37-41 kya come from? I could not find that in the supplement.

Minor comments

4.1 I would give the supplement another read as there are some missing references. For instance on page 5 you write "As such, we see substantial correlations between the highly related measures of fixed sites per individual, population-wise fixed sites and population-wise segregating sites (Fig S)." What figure is being referenced here?

The authors also write "Haplotype networks of putatively introgressed regions often show expected patterns (Fig. S12)," - what does often mean? 50% of the time, 90% of the time?

4.2 Figure S11 - could you please highlight the individual you are discussing? The font is very small and its hard to make out the relevant individuals like Gorilla_beringei_beringei-Bwiruka in panel A for instance.

Reviewer #3 (Remarks to the Author):

This manuscript focuses on introgression in gorillas from an extinct gorilla lineage. The approach used here replicates previous work on humans, the genus *Pan*, and other taxa. The authors find evidence of ghost admixture and identify regions of the genome that retain these introgressed tracts. This study provides another example of archaic admixture that increasingly appears to be the rule not the exception in the recent evolutionary history of mammals. Beyond evidence of archaic introgression, this manuscript largely recapitulates previous findings of gorilla demography where power allows. I think some additional interpretation and explanation of the selection results would be of interest to readers. Overall, the paper is well written and conceptually sound.

I list specific comments below. This version of the manuscript did not include line numbers so I include the relevant section title for each comment.

- Eastern gorillas form two population clusters/Data processing: How many variants are included in this analysis?
- Demographic modelling favours a ghost lineage in eastern gorillas: It may be worth noting here for readers less familiar with these methods that particular demographic events (e.g., ghost introgression into the gorilla common ancestor) are not easily inferred.
- Figure 2: I recommend using the units from panel A for the x-axes in panel B.

6- The ghost introgression landscape in eastern gorillas: What is the distribution of inferred lengths of these introgression tracts from hmix? How do they differ by subspecies?
- Figure 3: A dendrogram may be useful for panel A. Did the authors match for any characteristics of introgressed regions in the random regions aside from length in panel B? The grey box in panel E is difficult to see.
- The interaction of selection and introgression: This section was succinct and focused heavily on adaptive introgression. The authors note that genic regions were not depleted of archaic introgression. What about regulatory elements? Garcia-Pérez et al. 2021 recently annotated gorilla regulatory elements and these regions may exhibit depletion of archaic alleles as has been described for archaic variants in modern humans. It may also be worthwhile to further tease apart archaic introgression in genic regions by considering factors such as mutational tolerance per gene or conservation metrics.
- Figure 4: The authors may consider adding the adaptive introgression segments to this plot.
- Data processing: The authors note that they mapped raw reads to hg19 for comparable mapping bias. How often did reads fail to map to this reference? Certain regions of the gorilla genome may map poorly to a human reference and two recent high-quality gorilla reference genomes are available: gorGor5 and gorGor6.
- Detecting introgressed fragments: Why did the authors choose 40 kbp for simulating and identifying fragments with S*? Archaic fragments in modern Eurasians are approximately this size, on average and I would expect a similar size distribution for archaic tracts in gorillas.

*****END*****

Author Rebuttal to Initial comments

Responses to Reviewers

Reviewer #1 (Remarks to the Author):

This interesting manuscript by Pawar et al. focuses an important topic, i.e. ghost introgression, in evolutionary biology. Ghost introgression has been well documented in modern humans, in which ancient DNA from fossil remains are available, much facilitating the detection of introgression from Neanderthals and Denisovans. Using genomic data of extant species, researchers can also find the legacy of extinct species without fossil DNA. Applying the approach developed in their previous study of the Pan clade to gorilla species, the authors evaluated ghost introgression scenarios against what they called "null model" which involves gene flow only between extant gorilla species, and found the best model is ghost introgression into the common ancestor of two eastern gorilla lineages. Built on this model, they then identified the putative introgressed fragments in the genomes of eastern gorillas, particularly those fragments that are likely adaptive.

We thank the reviewer for the positive evaluation of our work.

However, I have some concerns with the results and conclusions in this manuscript. It is well known that the Approximate Bayesian Computation approach requires good prior knowledge for the models to be included in analysis, because there are numerous possible ghost introgression scenarios. For example, ghost lineages may contribute genetic materials to any one of 4 subspecies rather than any common ancestor. If I were you, I would first perform some exploratory phylogenomic analyses (e.g. ABBA-BABA test or 3s or PhyloNet/Phylonetworks) to narrow down possible gene flow scenarios. In other words, the two ghost introgression models should be suggested by data, rather than simply assumed as it appeared to me. At the very least, more background information and more clarification should be provided for the model selection.

We acknowledge that our demographic model, like any such model, is a simplification that serves to explain many specific features of the data. That means, indeed numerous possible ghost introgression scenarios exist, and we cannot rule out that further events exist beyond what we describe, especially with regards to much smaller amounts of ghost admixture. Given the possible complexity, we reason that it is unfeasible to explore the entire ghost parameter space using ABC (or other approaches available at this moment). Hence, we performed demographic modelling of biologically plausible scenarios, for which we could subsequently apply statistical methods to detect introgressed fragments in individual genomes (S* and hmix). The analysis of this introgression landscape lends further credibility to the scenario. For example, ghost lineages could theoretically have introgressed into both species' common ancestors, however this scenario is undetectable using these statistical methods. These statistical methods require an ingroup (which experienced the introgression) and an outgroup (which did not experience introgression). By applying two independent statistical methods designed to infer ghost introgression, we pursue an independent line of evidence additional to demographic modelling.

We agree that a ghost lineage could hypothetically have introgressed into any of the four subspecies. However, this is less plausible than introgression into the common ancestor of either group for several reasons. Firstly, the eastern gorilla divergence time is very shallow, estimated here at ~15kya (14.2-15.8kya 95% CI), and concordant with estimates from microsatellite data (Roy et al. 2014). Hence if a ghost had contributed gene flow to one of the eastern subspecies more recently than around 10kya, very long discordant haplotypes would likely have been detected in previous studies. Secondly, with the data currently available we do not have power to assess ghost gene flow only to Cross River gorillas, for which high-coverage, population-level whole genomes are not currently available. In principle ghost gene flow to western lowland gorillas is possible, but given the lack of signal of a defined ghost introgression pulse into the western common ancestor, we did not explore this further at the level of the western lowland subspecies. We explain this more explicitly now in the main text and supplementary material (S2 Exploratory phylogenomic analyses):

8

ESS
: is

Results: Nevertheless, unaccounted demographic events such as ancient population structure or ghost admixture could affect parameter estimates (Tricou et al. 2022), particularly given evidence in other great apes (Kuhlwilm et al. 2019). Initial exploratory analyses with f_4 -statistics and admixture graphs (SI 2) did not show any asymmetries between the four gorilla terminal populations, which would arise if ghost admixture had occurred in any of the individual subspecies. However, this does not exclude the possibility of ghost admixture into the common ancestor of eastern or western gorillas, which these methods cannot assess. To account for this and explicitly test if ghost admixture could improve the inferred null demographic model (model A), we considered two more complex demographic models..

Discussion: However, we note that further ghost introgression events may exist beyond what we describe, for example with regards to much smaller amounts of ghost admixture into gorillas, or with shallower divergence times of the ghost lineages, or in the context of larger effective population sizes in western gorillas.

It is indeed desirable to form hypotheses based on other methods. However, particularly for the case of ghost admixture and a topology of two clades, this is very challenging. We initially performed exploratory phylogenomic analyses, as suggested by the reviewer, using both f_4 -statistics and admixture graphs. If it were ghost admixture into any of the four terminal populations, these statistics might be informative, as asymmetries between the clades would be introduced, although this would still be confounded by possible gene flow events between the terminal populations. However, in our case, as explained above, the two testable and reasonable hypotheses involve the ancestral nodes of either western or eastern gorillas. Below we show the best fitting admixture graphs for 0 and 1 admixture edges, where the first node is a poorly defined signature of substructure in western lowland gorillas (0% proportion, see below). Further edges increase complexity across the subspecies, but we caution that increasingly complex graphs are less reliable (Maier et al. 2023). A graph based on the hypothesis of ghost admixture into the ancestors of eastern gorillas does provide a better fit to the f -statistics ($p=0.002$), although the best fitting graph with one edge does provide an even better fit ($p=0.002$). We caution that the best fitting admixture graphs may not represent the best possible scenario, as pointed out by the authors of the method (Maier et al. 2023). We have now added information on these analyses to the supplementary material (S2 Exploratory phylogenomic analyses). We note that these methods (as well as PhyloNet) are not designed to identify ghost lineages (Tricou, Tannier, and de Vienne 2022), as such we explicitly tailored our approaches to our question of interest.

Figure S1: Best fitting admixture graphs for 0 and 1 admixture edges for the four gorilla subspecies.

9

In order to reduce the complexity of the ghost introgression models in ABC, you “fixed parameters with narrow C_{is} from model A” (LINE 480). This is quite subjective, due to vague meaning of “narrow”. Is there any sound reason for the choice? By keeping most parameters of the null model as fixed in the two ghost introgression models, you implicitly make a big assumption that ghost introgression has little influence on these parameters. At least this assumption should be justified with sufficient details. To my mind, some parameters are apparently affected by the ghost introgression events, such as the effective population size of

ESS
: is

the common ancestor of all gorillas and the divergence time between eastern and western lineages, and they should be estimated in the models.

We thank the reviewer for raising these points. We restricted the parameter space of the ghost models by fixing the parameters which had been well inferred in the null model, acknowledging that “well-inferred” has a subjective aspect. To address this point, we have now re-performed parameter inference of the two ghost admixture models (into the common eastern ancestor, or into the common western ancestor) sampling all parameters from their priors (see Figure below). This analysis is now presented in detail in the supplementary material (S3.4 Revised demographic modelling, Figs. S11-S13, Table S3). Briefly, what we find is that a model with ghost admixture into the eastern ancestral population yields parameter estimates concordant with the model presented in the original manuscript, with 1.93% of introgression (0.33-2.45%, 95% CI). We again find that a ghost admixture event in the western ancestral population will be inferred very close to the species split time, suggestive of detecting population substructure in the western gorilla population rather than admixture. For both models, the posterior parameters are strongly correlated (eastern ghost models: $\rho=0.8531903$, $p=1.075e-06$; western ghost models $\rho=0.8870695$, $p=2.913e-06$). However, we note that due to the larger parameter space, the credible intervals of the posterior parameters are larger. When performing a model comparison, we still find the best support for the model of ghost admixture into the eastern ancestor as presented in the original manuscript (Bayes factor 823), with no accepted simulations for the two models with re-inferred parameters, even when using the best 10% (Table S5). We suggest that the widening of the priors when including ghost admixture comes at the cost of less precision across all parameters. Hence, we conclude that this demographic model is still the best approximation of gorilla history with the available data and tools at hand.

We briefly explain this in the main text:

Results: We assessed the robustness of our ghost models B and C using a wider parameter space (Fig S11-S13, see Methods), resulting in coherent posteriors with those observed in models B and C (Table S3), albeit with wider confidence intervals, as expected given the increased model complexity (Fig. S11).

Methods: To assess the impact of fixing well-inferred parameters from the null model on subsequent ghost parameter inference and explore the ghost parameter space more fully we undertook a revised modelling approach (Supplementary Material). In these revised ghost models, we performed parameter inference sampling all parameters from priors, for ghost gene flow into the common ancestor of D) eastern gorillas and E) western gorillas (Table S3). We observed a strong correlation between the estimated parameters of the original and the revised ghost models, albeit with wider posterior distributions for the revised models due to increased complexity and larger parameter space (Supplementary information)...We also performed cross validation and model comparison for the five demographic models: A) null demography, B) ghost gene flow into the eastern common ancestor, C) ghost gene flow into the western common ancestor, D) revised model of ghost gene flow into the eastern common ancestor and E) revised model of ghost gene flow into the western common ancestor, where we still observed model B having the highest support (Table S5, Supplementary Material).

Figure S11: Posterior distributions for the archaic introgression proportion, time of archaic introgression, and gorilla-ghost split time, for the revised models of ghost gene flow to **A** the common ancestor of eastern gorillas and **B** the common ancestor of western gorillas, sampling all parameters from priors. We note these are equivalent to Figure 2C, but for the revised ghost models (models D and E). The dotted line indicates the prior distribution. The black line indicates the posterior inferred with a simple 'rejection' algorithm. The red line represents the posterior inferred with neural networks. Distributions are plotted in ms units.

In the manuscript, you made no reference to Table S4-S8, and some references to supplementary Tables in the manuscript seem misplaced, causing much confusion and even misunderstanding. In your Fig. S4, the ghost introgression was placed in the common ancestor of all gorillas, not in the western common ancestor as claimed in the legend or main text (see LINE 189 and LINE 191).

11

We apologise for any confusion arising here. We have now re-arranged the tables, including merging posteriors and priors for better accessibility. The plot in Fig. S4 (now Fig. S9) refers to the results of the model of ghost gene flow to the western common ancestor, whereby the posteriors indicate a small contribution to the

ESS
: is

common ancestor of all gorillas (consistent with ancestral substructure), rather than a defined pulse to the western common ancestor. We have added this clarification to the figure legend in question.

LINE 454: The mutation rate is per year or per generation? Please specify.

The mutation rate given is per generation. We arrive at a mean of $1.235e-08$ as follows. In Figure 3 of Besenbacher et al. (Besenbacher et al. 2019) a value of 0.65 is given for the absolute mutation rate per billion years for gorillas. We divide 0.65 by 1000000000 and multiply by a generation time of 19 years to get a value for the mutation rate per generation. We now specify this in the methods (Demographic modelling: null demographic model section).

Reviewer #2 (Remarks to the Author):

Pawar et al sequence new eastern gorillas and model the demographic history of all gorilla sub-species. They then check if there is any evidence of ghost admixture into any of the common ancestors of western and eastern gorillas. They conclude that a demographic model which incorporates ghost admixture into eastern gorillas fits their summary statistics better than a model without. They run two methods developed for detecting archaic introgression into modern humans. They find evidence that a deeply divergent ghost population contributed 2-3% to the genomes of eastern gorillas.

The findings are well presented and I was able to find most relevant information in the supplementary material. The authors have done a lot of work to produce these results and they provide links to the scripts in the supplementary tables which is good for rerunning analysis in the future and make the science reproducible. Good job!

The findings are interesting and I find it plausible that admixture from a ghost population into eastern gorillas took place as hybridization events are common in mammals. However I would like to see some additional analysis to ensure that these findings are robust.

We thank the reviewer for their positive evaluation of our work and for the appreciation of our efforts for open and reproducible science.

Simulations

Establishing a demographic model for Gorilla species is key for this manuscript and needs some additional work.

1.1 The authors provide a detailed overview in the supplement of all their analysis and there are a lot of tabs to go through plus the naming is not intuitive. For example it is hard to remember what "e_moun_t3.1" means and for "gor_ghost_anc" the explanation is "split of all gorillas + ghost population" but the units are in individuals so it is an effective population size?

I think it could be useful to merge the tabs with priors and posteriors and have the first column be "Explanation column". I have attached a suggestion of what it could look like as "example_supp.xlsx"

12

This is indeed an important point, and we agree that an improvement of the supplementary tables as suggested will help the reader. We have now implemented a new supplementary table with the different priors and posteriors of the different scenarios, better naming and structure (Table S3). We now also summarise the core models - the best inferred demographic model (model B, with ghost gene flow to the eastern common ancestor) and the null model (model A, without archaic introgression) in Table S2.

ESS
: is

1.2 I understand that the search space becomes very large so some parameters need to be fixed but it feels like the authors might be getting at a local optimum and not a global one. For instance in tab "8.ghostw.posterior" only 7 parameters are estimated while the others are fixed. For instance the parameter "w_low_t0" is set to 64.7488. However if I look in "4.null.posterior" this parameter also has uncertainty (23.758 - 95.9855). What happens if you use this range as a prior when fitting parameters for ghost admixture?

We thank the reviewer for their comments. A similar point was raised by another reviewer, and we acknowledge that models need to be shown as robust as possible. Hence, we now explore the parameter space more fully, and mitigate the chance of getting at a local rather than global optimum. In order to do this, we have re-performed parameter inference for the ghost models sampling all parameters from the same range of priors. Since these are the same as those used for the null model, this includes the 95% credible intervals. We also document the inference and results in the supplementary materials. Briefly, we have re-performed parameter inference of the two ghost admixture models (into the common eastern ancestor, or into the common western ancestor) sampling all parameters from their priors. A model with ghost admixture into the eastern ancestral population yields parameter estimates concordant with the model presented in the original manuscript, with 1.93% of introgression (0.33-2.45%, 95% CI). We again find that a ghost admixture event in the western ancestral population will be inferred very close to the species split time, suggestive of detecting population substructure in the western gorilla population rather than admixture. For both models, the posterior parameters are strongly correlated (eastern ghost models: $\rho=0.8531903$, $p=1.075e-06$; western ghost models $\rho=0.8870695$, $p=2.913e-06$). However, we note that due to the larger parameter space, the credible intervals of the posterior parameters are larger. When performing a model comparison, we still find the best support for the model of ghost admixture into the eastern ancestor as presented in the original manuscript (Bayes factor 823), with no accepted simulations for the two models with re-inferred parameters, even when using the best 10%. We suggest that the widening of the priors when including ghost admixture comes at the cost of less precision across all parameters. Hence, we conclude that this demographic model is still the best approximation of gorilla history with the available data and tools at hand. This analysis is now presented in detail in the supplementary material (S3.4 Revised demographic modelling, Figs. S11-S13, Table S3) and briefly mentioned in the main text:

Results: We assessed the robustness of our ghost models B and C using a wider parameter space (Fig S11-S13, see Methods), resulting in coherent posteriors with those observed in models B and C (Table S3), albeit with wider confidence intervals, as expected given the increased model complexity (Fig. S11).

Methods: To assess the impact of fixing well-inferred parameters from the null model on subsequent ghost parameter inference and explore the ghost parameter space more fully we undertook a revised modelling approach (Supplementary Material). In these revised ghost models, we performed parameter inference sampling all parameters from priors, for ghost gene flow into the common ancestor of D) eastern gorillas and E) western gorillas (Table S3). We observed a strong correlation between the estimated parameters of the original and the revised ghost models, albeit with wider posterior distributions for the revised models due to increased complexity and larger parameter space (Supplementary information)... We also performed cross validation and model comparison for the five demographic models: A) null demography, B) ghost gene flow into the eastern common ancestor, C) ghost gene flow into the western common ancestor, D) revised model of ghost gene flow into the eastern common ancestor and E) revised model of ghost gene flow into the western common ancestor, where we still observed model B having the highest support (Table S5, Supplementary Material).

13

1.3 Furthermore some parameters are always fixed from the beginning t1, t2, t3. But t4 in 2.suppl.initial.mergedsimns (5. gorilla split time (low divergence simulations) is allowed to vary. The estimate value is 965,481 years ago which is very different from previous estimates of 261,000 years ago (McManus et al. 2015) and 429,000 years ago (Scally et al. 2012). Can the authors provide a brief explanation for why some parameters are fixed while some are not?

ESS
: is

Indeed, there is an explanation for this point: the parameters t_1 - t_3 are fixed in the null model, since these are very recent events, with priors from the literature that were very narrow. The summary statistics used here have little power to determine posteriors in this setting and indeed, in initial iterations of ABC-based modelling, we observed that these were contributing noise, but would contribute little information to the question of deeper demographic history. Hence we decided to fix these parameters at the midpoint of their prior ranges to reduce complexity of the model and focus on the history further back in time.

The parameter t_6 , pertaining to the time of extant admixture (between western lowland gorillas and the common eastern ancestor) is also fixed in the null model at 34 kya. The Generalised Phylogenetic Coalescent Sampler (G-PhoCS) modelling used by McManus et al. (McManus et al. 2015) implemented migration throughout the entire simulation. To convert this continuous migration to migration pulses we took the midpoint between the WLG-CRG split time inferred by McManus et al. (McManus et al. 2015) and the present ($68/2=34$ kya) as the timing for extant admixture.

All other parameters are allowed to vary, sampling from priors informed by previous literature. We have added this information to the supplementary material (S3.3 Adjusted demographic modelling).

Regarding the deep divergence time of the two species, it is the case that in this study we infer deeper divergence times. However, based on initial models, a more shallow divergence time in simulations did not match well the observed genomic patterns, such as segregating site distributions in the real data. We admit that in this study we cannot explore the full complexity of the ancestral population, including a prolonged separation process. However, it is important to note that the expected S^* scores increase with divergence time, and using such a deep divergence time is a conservative approach to detect archaic introgression within the context of this study.

----- Archiac introgression

This is the section which is most crucial to the point the authors want to make. First all models are approximations - for instance you model the effective population size of Eastern gorillas as being a constant 5,325 for almost a million years. This is of course an approximation. When one adds a ghost admixture one allows some extra flexibility in the model so scenarios with ghost admixture almost always fit the data better because there will always be some deeply divergent haplotypes present. So we have to make sure that this ghost admixture is robust to model mis-specifications.

2.1) I suggest creating msprime simulated data using the demographic model from Fig. 2 - (extended table S10). The advantage of using msprime is that you can keep track of the introgressed segments and evaluate how well you are doing in identifying them! But I would pick the values which increases the amount of deeply divergent haplotypes due to ILS and see if you can still distinguish that scenario from one with ghost admixture. This would mean increasing all effective population size estimates of ancestral populations to the maximum value of their credibility intervals e.g. increase the population size of Eastern gorillas to 23015.4 and the parameter `ne_gor_species_split` to 23298.3. Basically every population size that is before the admixture event. Let refer to this scenario as "Scenario_X" which represents a worst case scenario for identifying ghost admixture when using sstar or hmmix.

Then you could add ghost admixture of 2.5% at 38,000 years ago and let's call this scenario "Scenario_Y".

2.2) Now if you run hmmix or sstar (whichever one is easiest to run) on both scenarios using the same approach as you describe in the supplement how much "ghost admixture" does the model identify in ScenarioX vs ScenarioY?

2.3) For ScenarioY where you know which segments are actually introgressed what is the false positive, false negative rates of identifying these segments? It would also be helpful to show the length distribution of 'ghost admixture' segments for scenarioX, scenarioY and what you observe in real data like you do in Figure S2.

14

ESS
: is

We thank the reviewer for these valuable suggestions, which we have endeavoured to implement. We include this assessment of the robustness of the S^* statistic and *hmmix* to misspecifications of the demographic model in a new supplementary section S3.6 Validation of method performance. As suggested by the reviewer we generated simulations using *msprime* (Baumdicker et al. 2022; Kelleher, Etheridge, and McVean 2016) under different null demographic models and assessed the performance of the S^* statistic and *hmmix* using precision-recall curves (Fig. S14). We first generated a generalized additive model of the expected distributions of S^* scores, as described for the main analysis in the manuscript, for both the original null model (model A in the manuscript using the weighted median posteriors) and a modified null model where we take the maximum value of the 95% credible interval for all ancestral N_e parameters (here termed “worst” null model).

We then simulated data on the model of archaic introgression as presented in the manuscript (model B), as well as a modified model of archaic introgression with the maximum values of the 95% credible interval for all ancestral N_e parameters (“worst” model). We subsequently run S^* and *hmmix*, with a range of values for the quantile (threshold to define outliers of the statistic) of 0-0.999 for the S^* statistic and posterior probabilities of 0-0.9999 for *hmmix*, following (Huang et al. 2022) (Tables S8-S9). We simulated 10 individuals of western lowland gorillas as outgroup, and 1 individual for eastern lowland or mountain gorillas, respectively, as target individuals. We performed 100 replicates of each model. For the S^* statistic, we then explored a “worst mis-specified” scenario, where we generated simulated data under the “worst” model (with likely high ILS), but run the S^* analysis using the expected S^* scores for model A (expecting less ILS) (Fig. S14A).

The performance of the S^* statistic under model A exhibits good precision and recall, with a 90.96% detection rate of true introgressed fragments for eastern lowland gorillas (91.06% for mountain gorillas) at the 99% quantile (Fig. 2B, S14, Table S8), which is comparable to the human-Neanderthal scenario (Huang et al. 2022). Whereas, under model misspecification with a simulated inflated contribution of ILS, the recall of S^* remains high but the precision falls to 55.82% for eastern lowland gorillas (53.33% for mountain gorillas), since the false discovery rate increases as expected. We conclude that the S^* statistic performs well in detecting introgressed fragments under our null model (model A), even if the true demography was deviating in terms of ancestral effective population sizes. We added this information into the main text:

Results: We assessed the performance of the S^ statistic using coalescent simulations where we could trace the introgressed fragments (Methods). The precision and recall are high, with a 90.96% detection rate of true introgressed fragments for eastern lowland gorillas (91.06% for mountain gorillas) at the 99% quantile (Fig. 2B, S14, Table S8, see Methods), comparable to the human-Neanderthal scenario (Huang et al. 2022). Since the CIs of the null demographic model encompass larger effective population sizes, which would lead to inflated rates of incomplete lineage sorting that might affect the expected distribution of S^* scores, we also assessed how these parameters influence our findings. Using the maximum values within the 95% CIs, we find that the recall of the S^* statistic remains high, while the precision falls to 55.82% for eastern lowland gorillas (53.33% for mountain gorillas), reflecting an increase in the false discovery rate, as expected. We conclude that the S^* statistic performed well in detecting introgressed fragments under our null model, even when assuming misspecification of the null model.*

*Analogous to previous work (Kuhlwilm et al. 2019), we also employed *hmmix* to detect introgressed windows (Skov et al. 2018), which performs well for the given demographic model (Fig. 2B), with precision and recall well above 80% (Table S9).*

Figure S14: Precision-recall curves under different null demographic scenarios for the S^* statistic at the 99% quantile as implemented in *sstar* (Huang et al. 2022) and *hmmix* at the 95% posterior probability cutoff. Main model refers to a model taking the weighted median posteriors from the ABC-based null demography presented herein (Fig. S6A). Worst model refers to a model taking the maximum value of the 95% credible interval for all ancestral N_e parameters from the ABC-based null demography. For the S^* statistic, we consider the target population as alternately eastern lowland or mountain gorillas, e.g. Main Model EL. Worst mis-specified is where we generate simulated data under the worst model but run the S^* analysis using the ‘quantile’ or outlier values inferred under the main model. Skov=*hmmix* method, EL=eastern lowland gorillas, M=mountain gorillas.

Panel 2B Precision and recall of *hmmix* (at the 95% posterior probability cutoff) and the S^* statistic (at the 99% quantile using *sstar* (Huang et al. 2022)) in simulated data using *msprime*. Precision (percentage of recovered introgressed fragments) and recall (percentage of true among inferred introgressed fragments) for *hmmix* and S^* (for ELG = eastern lowland gorilla and MG = mountain gorilla). Dark bars represent performance using the model presented in panel A, light bars represent the “worst” model with large effective population sizes, in the case of *hmmix* to simulate the data to detect fragments, in the case of S^* to obtain the expected distribution of S^* scores.

Additional questions

3.1 Further I was wondering why you don't train the parameters using *hmmix*. This should give you the coalescent time to the outgroup for 'gorilla' and 'ghost' segments. Can you provide the trained parameters? Are these consistent with the one you infer using ABC?

We apologize for omitting this aspect of *hmmix*. The parameter training was performed for *hmmix*, which is important for decoding as well. When writing the manuscript, we did not include a table of trained parameters, which we do now in Table S11 and further detail in S3.5 Parameters from *hmmix*. We find that the coalescence times between the two gorilla species are inferred at ~256 kya, which is more recent than the estimates from the ABC modelling. However, we would like to point out that reversing ingroup and outgroup (i.e. using western lowland gorillas as potential ingroup and eastern gorillas as potential outgroup) yields a larger coalescence time of ~572 kya due to the larger effective population size. This relationship between population size and coalescence time makes it difficult to compare to the divergence times of the ABC modelling, especially with a method that was developed for the human-Neanderthal scenario. Furthermore, we infer a coalescence time of gorilla and ghost segments at 1.520 Mya, but also an archaic percentage of 17.7%, which is a larger archaic proportion at a shallower coalescence. The calculated admixture time is ~69 kya, hence older than the one inferred in the demographic model as well. We suggest that this may reflect a more complex history in the deep past of the gorilla populations than represented in the model, possibly with several events of migration or substructure within eastern gorillas. This will need to be explored further in future studies. We apply thorough filtering for decoding the introgressed fragments with *hmmix*, leading to a largely overlapping set of candidate regions with those inferred with S^* , as shown in the manuscript. A more lenient filtering would give larger numbers of segments which may, however, not reflect the same ghost introgression event.

3.2. Why are you mapping Gorilla shotgun reads to the human reference hg19 - why not use the gorilla reference genome (gorgor6 is the latest version I believe).

This approach does indeed have a reasoning in avoiding reference bias. In our experience, using a reference genome of one population leads to a bias in calling variants from the other population. The gorilla reference genome was assembled from a western gorilla individual. For example, when using the western chimpanzee reference genome, ABBA/BABA statistics with central chimpanzees and bonobos suggested an exaggerated amount of allele sharing (de Manuel et al. 2016), while using the human reference genome provided an unbiased estimate. Even though this case of ghost admixture does not rely on allele sharing, we decided to avoid such potential effects. We have added a brief explanation to this effect in the Methods (Data processing section). Furthermore, functional annotations are more complete based on the human genome, which is important for downstream analyses. In this case, using the gorilla reference genome would require another complication from leftover of variants, which we also try to avoid.

3.3. You refer to S^* and *hmmix* as being "two complementary approaches". However the main signal for both of them is SNP density of variants not found in an outgroup and the fact that these variants are somewhat clustered. So it is not surprising that they overlap! I would make that clear to the reader.

We apologise for any confusion arising. Our intended meaning was to highlight that although S^* and *hmmix* target the same signature (of introgression from an unsampled lineage), the algorithms are independent and distinct. Most patently, S^* requires the inclusion of a null demographic model in order to determine outliers of the statistic in windows of a given size (in this case, 40 kbp, which may represent the expected size of introgressed fragments), whereas *hmmix* assigns regions as 'internal' or 'external' based on the density of private SNPs without demographic information in very small regions (in this case, 1 kbp). We now make this more clear by adding modifying the statement to:

Results: we implemented two *independent* approaches: the S^* statistic (Plagnol and Wall 2006; Vernot and Akey 2014) and the SkovHMM method, or *hmmix* (Skov et al. 2018)...Hence, although both S^* and *hmmix* target the same signature of ghost introgression, the algorithms are distinct.

17

ess
is

3.4. The authors write: "We find that 1.48-2.97% of the individual eastern gorilla genomes are inferred as external at a strict threshold for the mean probability of 0.95, with an estimated introgression time of 37-41 kya." Where does the 37-41 kya come from? I could not find that in the supplement.

We agree that it was not clear that this information was obtained from the median length of introgressed regions obtained with hmmix. We now clarify this in the main text, and provide more details in Table S11.

Minor comments

4.1 I would give the supplement another read as there are some missing references. For instance on page 5 you write "As such, we see substantial correlations between the highly related measures of fixed sites per individual, population-wise fixed sites and population-wise segregating sites (Fig. S)." What figure is being referenced here?

We apologise for this omission, we intended to refer to Fig. S3A (which is now Fig. S5A) and have now added this.

The authors also write "Haplotype networks of putatively introgressed regions often show expected patterns (Fig. S12)," - what does often mean? 50% of the time, 90% of the time?

4.2 Figure S11 - could you please highlight the individual you are discussing? The font is very small and its hard to make out the relevant individuals like *Gorilla_beringei_beringei*-Bwiruka in panel A for instance.

We thank the reviewer for pointing out these minor points. We fixed references in the supplementary material, and improved Figure S11 (now Fig. S18). Regarding the haplotype networks, this proportion is 90% among the 20 longest loci (i.e. for the longest 20 introgressed regions 18 haplotypes look archaic in origin). However, this is difficult to assess manually for the full number of loci.

Updated Figure S18: A NJ tree of SNPs in all putative introgressed regions of mountain gorilla individual 1 (*Gorilla beringei beringei-Bwiruka*) and of random genomic regions of equivalent length distribution. **B** NJ tree of SNPs in putative introgressed regions unique to mountain gorilla individual 1 (so-called 'private introgressed regions') and equivalent random genomic regions. The target individual is indicated by a red star in each case.

Reviewer #3 (Remarks to the Author):

This manuscript focuses on introgression in gorillas from an extinct gorilla lineage. The approach used here replicates previous work on humans, the genus *Pan*, and other taxa. The authors find evidence of ghost admixture and identify regions of the genome that retain these introgressed tracts. This study provides another example of archaic admixture that increasingly appears to be the rule not the exception in the recent evolutionary history of mammals. Beyond evidence of archaic introgression, this manuscript largely recapitulates previous findings of gorilla demography where power allows. I think some additional interpretation and explanation of the selection results would be of interest to readers. Overall, the paper is well written and conceptually sound.

We thank the reviewer for their overall positive assessment of our manuscript. We improved the manuscript based on the comments, specifically with additional analyses on the consequences of this admixture event, as explained point by point below.

I list specific comments below. This version of the manuscript did not include line numbers so I include the relevant section title for each comment.

19

355
15

- Eastern gorillas form two population clusters/Data processing: How many variants are included in this analysis?

We now provide the numbers of variants in the methods 'Detecting introgressed fragments' section: *For the S* analysis 15,181,832 variants were included.*

- Demographic modelling favours a ghost lineage in eastern gorillas: It may be worth noting here for readers less familiar with these methods that particular demographic events (e.g., ghost introgression into the gorilla common ancestor) are not easily inferred.

We agree with the reviewer that not all possible scenarios can be explored or explained by our analysis. We are now more explicit on this in the main text:

Discussion: *However, we note that further ghost introgression events may exist beyond what we describe, for example with regards to much smaller amounts of ghost admixture into gorillas, or with shallower divergence times of the ghost lineages, or in the context of larger effective population sizes in western gorillas.*

- Figure 2: I recommend using the units from panel A for the x-axes in panel B.

We agree that using real units is more meaningful for this figure and have improved it accordingly.

Updated panel 2C (previously 2B) Posterior distributions for the archaic introgression proportion, time of archaic introgression, and gorilla-ghost split time. The dotted line indicates the prior distribution. The black line indicates the posterior inferred with a simple 'rejection' algorithm. The red line represents the posterior inferred with neural networks. Compared to the rejection algorithm, neural networks reduce the dimensionality of the summary statistics used and account for possible mismatch between the observed and simulated summary statistics (Csilléry, François, and Blum 2012).

- The ghost introgression landscape in eastern gorillas: What is the distribution of inferred lengths of these introgression tracts from hmix? How do they differ by subspecies?

20

ESS
: is

We do not observe a significant difference between the subspecies in the distribution of introgressed tract lengths inferred from hmmix ($p > 0.01$, Wilcoxon unpaired test for both 0.9 and 0.95 threshold), as we now show in Fig. S16:

Figure S16: Distribution of hmmix fragment lengths for mountain gorillas (blue) and eastern lowland gorillas (green) at a threshold of **A** 0.9 and **B** 0.95.

- Figure 3: A dendrogram may be useful for panel A. Did the authors match for any characteristics of introgressed regions in the random regions aside from length in panel B? The grey box in panel E is difficult to see.

We thank the reviewer for pointing out these aspects, and have improved the figure accordingly, by adding the dendrogram and modifying the colour (updated version of Figure 3 below). The introgressed regions were matched for length and proportion of positions with sufficient coverage of callable sites within the region in order to avoid regions without callable sites. This is now explained in the figure legend.

Updated Figure 3: Characterization of introgressed fragments. **A** Sharing of putative introgressed regions across eastern gorillas for autosomal regions detected using the S^* statistic and hmix. **B** Pairwise nucleotide differences in introgressed regions (x axis) and in random regions (y axis) matched for length and proportion of positions with sufficient coverage (i.e. avoiding genomic regions without callable sites). Colours indicate the comparison: among eastern gorillas (EG-EG, green), among western gorillas (WG-WG, orange), and between eastern and western gorillas (EG-WG, purple). **C** Percentage of overlapping base pairs in introgressed regions (red lines) and random regions (violin plots) for eastern gorillas. For details on the definition of random regions see Methods. **D** Percentage of protein coding content detected in introgressed regions (red lines) and random regions (violin plots) for eastern gorillas. **E** Percentage of high impact GERP content detected in introgressed regions (red lines) and random regions (violin plots) for eastern gorillas. **F** Autosome:X ratio of introgressed fragments inferred using hmix for eastern gorillas (violin plots), with reference lines for the equivalent values for bonobos (red line) and humans (distribution as grey bar). In panels C-F, MG = mountain gorillas, EL = eastern lowlands.

- **The interaction of selection and introgression:** This section was succinct and focused heavily on adaptive introgression. The authors note that genic regions were not depleted of archaic introgression. What about regulatory elements? Garcia-Pérez et al. 2021 recently annotated gorilla regulatory elements and these regions may exhibit depletion of archaic alleles as has been described for archaic variants in modern humans. It may also be worthwhile to further tease apart archaic introgression in genic regions by considering factors such as mutational tolerance per gene or conservation metrics.

22

We thank the reviewer for their valuable suggestions. We agree that a more comprehensive study of functional effects is of interest here, and undertook an investigation of regulatory elements in introgressed fragments, detailed in S4.4 Functional consequences: regulatory elements. We assessed the proportion of regulatory base pairs within putative introgressed and random regions of equivalent length and callability, using the gorilla-defined regulatory element annotations of García-Pérez et al. (García-Pérez et al. 2021). We assess this both from a global perspective and per regulatory element type (poised, strong, weak, enhancers

ESS
: is

and promoters). We note that this data derives from gorilla lymphoblastoid cells (LCLs), which means that most of the patterns of expression may be cell-type dependent and specific regulatory effects, for example during brain development, would not be recovered. This is an inherent limitation of this kind of analysis in a non-human context. Moreover, the two gorilla LCL replicates belonged to the western species, hence are equidistant to both eastern subspecies.

We find no difference in the overall proportion of regulatory base pairs in putative introgressed regions compared to random genomic regions for either eastern gorilla population (Figure below, Fig. S24A). However, when we consider the proportion of regulatory base pairs per regulatory element we see an excess of strong enhancers (sE) in mountain gorilla introgressed regions, compared to random regions (Figure below, Fig. S24B). These sE are largely intragenic enhancers (Figure below, Fig. S25), which agrees with patterns of regulatory architecture observed in primate sE more generally by (García-Pérez et al. 2021).

Furthermore, García-Pérez et al. (García-Pérez et al. 2021) had annotated which genes are associated with each regulatory element. Taking these annotations and filtering to genes with one-to-one orthologs across the primates considered by García-Pérez et al. (García-Pérez et al. 2021) (humans, chimpanzees, gorillas, orangutans and macaques) we define two sets of candidate genes: 1) genes regulated by sE in mountain gorilla introgressed regions (235 genes), and 2) genes regulated by sE in mountain gorilla adaptively introgressed regions (45 genes). We performed an over-representation analysis of our candidate genes for gene ontology terms using the WebGestaltR package and default settings (Liao et al. 2019). Our background set consisted of genes regulated by gorilla sE (again taking those genes with one-to-one orthologs in primates). No gene ontology category reached the significance threshold of FDR=0.05 with Benjamini-Hochberg correction (Figure below, Fig. S26). The top gene ontology categories detected relate to the LCL cell type, namely 'establishment of lymphocyte polarity' (p-value=0.00018256, FDR=0.38985) and 'establishment of T cell polarity' (p-value=0.00018256, FDR=0.38985) for candidate gene set 1) and 'forebrain generation of neurons' (p-value=0.000066791, FDR=0.14263) for candidate gene set 2).

Figure S24: Proportion of regulatory base pairs in introgressed regions (red lines) and random regions (violin plots) population wide in **A** and per regulatory element type for **B** mountain gorillas and **C** eastern lowland gorillas. Abbreviations represent: pE=poised enhancer, pP=poised promoter, sE=strong enhancer, sP=strong promoter, wE=weak enhancer, wP=weak promoter.

Figure S25: Gene regulatory architecture of strong enhancers in mountain gorilla introgressed regions (red points) and random genomic regions of equivalent length and callability (violin plots). Abbreviations represent: EiE=enhancer-interacting enhancer, gE=intragenic enhancer, gP=genic promoter, PiE=promoter-interacting enhancer, prE=proximal enhancer. This analysis only considers those strong enhancers which could be annotated to genes by Garcia-Pérez et al. (García-Pérez et al. 2021).

Figure S26: Over-representation in gene ontology categories for **A** genes regulated by sE in mountain gorilla introgressed regions and **B** genes regulated by sE in mountain gorilla adaptively introgressed regions. No category reaches significance at FDR=0.05 with Benjamini-Hochberg correction.

To address the question of mutational tolerance, specifically whether more deleterious mutations are observed in introgressed rather than random genomic regions, we assessed different measures of deleteriousness: genomic evolutionary rate profiling (GERP), SIFT, PolyPhen-2 and LINSIGHT scores (Davydov et al. 2010; Kumar, Henikoff, and Ng 2009; Adzhubei et al. 2010; Huang, Gulko, and Siepel 2017). We downloaded the pre-computed base-wise GERP scores for hg19 (Davydov et al. 2010) and considered sites (>4) as high impact and ($-2 < x < 2$) as low impact. SIFT and PolyPhen-2 scores were extracted from VEP annotation for missense variants. We consider sites annotated with (SIFT='deleterious' or 'deleterious_low_confidence'; PolyPhen-2='probably_damaging' or 'possibly_damaging') as high impact and (SIFT='tolerated' or 'tolerated_low_confidence'; PolyPhen-2='benign') as low impact. LINSIGHT scores incorporate epigenomic information, including chromatin accessibility and transcription factor binding (Huang, Gulko, and Siepel 2017). We downloaded the pre-calculated LINSIGHT scores for hg19 (Huang, Gulko, and Siepel 2017). This is now detailed in the supplementary material S4.3 Functional consequences: mutational tolerance.

For GERP, SIFT and PolyPhen-2 scores we calculated the proportion of high impact sites within putative introgressed regions and random regions as defined above. We calculated the mean LINSIGHT score across regions, since few high impact sites (>0.8) were identified in our dataset. We find a higher proportion of high impact GERP sites in introgressed regions of eastern lowland gorillas compared to mountain gorillas (panel E of the updated Fig. 3 above). However, for SIFT, PolyPhen-2 and LINSIGHT scores the introgressed regions of both eastern lowland and mountain gorillas follow random expectation (Figure below, Fig. S23).

Figure S23: Mutational conservation in introgressed fragments. Proportion of high impact sites in introgressed regions (red lines) and random regions (violin plots) for **A** SIFT scores and **B** PolyPhen-2 scores. High impact sites are those annotated as 'deleterious' and 'deleterious low confidence' for SIFT, and 'probably damaging' and 'possibly damaging' for PolyPhen-2. **C** Mean LINSIGHT score across introgressed regions (red lines) and random regions (violin plots).

25

• Figure 4: The authors may consider adding the adaptive introgression segments to this plot.

We initially did include these segments, however, in our opinion it is rather confusing, as the genes with signatures of adaptive introgression consist of short segments, and the overall landscape is presented on megabase-scale calculations. Hence, we decided to not include them in the manuscript. Below, we provide the

ess
is

version with this annotation below. We would change it in the manuscript if the reviewer suggests that it is still informative.

Figure 4 with adaptive introgressed candidate genes. Distribution of introgressed fragments. Outer circle: karyogram of the autosomes based on the human genome (hg19). Second circle from outside: Introgression landscape in mountain gorillas (blue), as cumulative amount of introgressed material in sliding windows of 2 million base pairs (Mbp). Third circle from outside: Introgression landscape in eastern lowland gorillas (green) in sliding windows of 2 Mbp. Inner circle: long regions depleted of introgression content are shown in orange (length ≥ 5 Mbp) and red (length ≥ 8 Mb). Candidate genes with signatures of adaptive introgression are labelled. Grey: Genomic regions with sufficient data ($>20\%$ of 40 kbp windows passing threshold). White: Genomic regions without sufficient data.

- **Data processing:** The authors note that they mapped raw reads to hg19 for comparable mapping bias. How often did reads fail to map to this reference? Certain regions of the gorilla genome may map poorly to a human reference and two recent high-quality gorilla reference genomes are available: gorGor5 and gorGor6.

We agree that it would be interesting to study the differential amounts of mapping to different reference genomes, which would be beyond the scope of this study. The reason for using the human reference genome in this particular case is actually to avoid reference bias. In our experience, using a reference genome of one population leads to a bias in calling variants from the other population. The gorilla reference genome was assembled from a western gorilla individual. For example, when using the western chimpanzee reference genome, ABBA/BABA statistics with central chimpanzees and bonobos suggested an exaggerated amount of allele sharing (de Manuel et al. 2016), while using the human reference genome provided an unbiased estimate. Even though this case of ghost admixture does not rely on allele sharing, we decided to avoid such potential effects. We have added a brief explanation to this effect in the Methods (Data processing section).

• Detecting introgressed fragments: Why did the authors choose 40 kbp for simulating and identifying fragments with S^* ? Archaic fragments in modern Eurasians are approximately this size, on average and I would expect a similar size distribution for archaic tracts in gorillas.

We admit that the choice of window length is somewhat arbitrary, and based on previous studies where S^* was successfully implemented for this purpose. In studies on modern humans (Vernot et al. 2016), a window size of 50 kbp was used, in a previous study on bonobos (Kuhlwilm et al. 2019), of 40 kbp. In principle, shorter windows may capture older events (such as in bonobos), while losing power in cases of species with low diversity (such as humans). As we originally did not have an a priori hypothesis on the timing of introgression of a possible ghost introgression event, we chose a somewhat smaller window size, which then formed the base of the demographic modelling with ABC. Exploring this tradeoff fully in different demographic scenarios (not only for the case of gorillas) will be part of a future study to assess the performance of different tools, but is beyond the scope of this study. In a performance assessment of the methods which we have now added to the manuscript, we used windows for 40 kbp, finding that we can recover ~91% of introgressed fragments in simulated data.

References

- Adzhubei, Ivan A., Steffen Schmidt, Leonid Peshkin, Vasily E. Ramensky, Anna Gerasimova, Peer Bork, Alexey S. Kondrashov, and Shamil R. Sunyaev. 2010. "A Method and Server for Predicting Damaging Missense Mutations." *Nature Methods* 7 (4): 248–49.
- Baumdicker, Franz, Gertjan Bisschop, Daniel Goldstein, Graham Gower, Aaron P. Ragsdale, Georgia Tsambos, Sha Zhu, et al. 2022. "Efficient Ancestry and Mutation Simulation with Msprime 1.0." *Genetics* 220 (3).
- Besenbacher, Søren, Christina Hvilsom, Tomas Marques-Bonet, Thomas Mailund, and Mikkel Heide Schierup. 2019. "Direct Estimation of Mutations in Great Apes Reconciles Phylogenetic Dating." *Nature Ecology & Evolution* 3 (2): 286–92.
- Csilléry, Katalin, Olivier François, and Michael G. B. Blum. 2012. "Abc: An R Package for Approximate Bayesian Computation (ABC)." *Methods in Ecology and Evolution*. <https://doi.org/10.1111/j.2041-210x.2011.00179.x>.
- Davydov, Eugene V., David L. Goode, Marina Sirota, Gregory M. Cooper, Arend Sidow, and Serafim Batzoglou. 2010. "Identifying a High Fraction of the Human Genome to Be under Selective Constraint Using GERP++." *PLoS Computational Biology* 6 (12): e1001025.
- García-Pérez, Raquel, Paula Esteller-Cucala, Glòria Mas, Irene Lobón, Valerio Di Carlo, Meritxell Riera, Martin Kuhlwilm, et al. 2021. "Epigenomic Profiling of Primate Lymphoblastoid Cell Lines Reveals the Evolutionary Patterns of Epigenetic Activities in Gene Regulatory Architectures." *Nature Communications* 12 (1): 3116.
- Huang, Yi-Fei, Brad Gulko, and Adam Siepel. 2017. "Fast, Scalable Prediction of Deleterious Noncoding Variants from Functional and Population Genomic Data." *Nature Genetics* 49 (4): 618–24.
- Kelleher, Jerome, Alison M. Etheridge, and Gilean McVean. 2016. "Efficient Coalescent Simulation and Genealogical Analysis for Large Sample Sizes." *PLoS Computational Biology* 12 (5): e1004842.
- Kuhlwilm, Martin, Sojung Han, Vitor C. Sousa, Laurent Excoffier, and Tomas Marques-Bonet. 2019. "Ancient Admixture from an Extinct Ape Lineage into Bonobos." *Nature Ecology & Evolution* 3 (6): 957–65.
- Kumar, Prateek, Steven Henikoff, and Pauline C. Ng. 2009. "Predicting the Effects of Coding Non-Synonymous Variants on Protein Function Using the SIFT Algorithm." *Nature Protocols* 4 (7): 1073–81.
- Liao, Yuxing, Jing Wang, Eric J. Jaehnig, Zhiao Shi, and Bing Zhang. 2019. "WebGestalt 2019: Gene Set Analysis Toolkit with Revamped UIs and APIs." *Nucleic Acids Research* 47 (W1): W199–205.
- Maier, Robert, Pavel Flegontov, Olga Flegontova, Ulas Ilisdak, Piya Changmai, and David Reich. 2023. "On the Limits of Fitting Complex Models of Population History to -Statistics." *eLife* 12 (April). <https://doi.org/10.7554/eLife.85492>.
- Manuel, Marc de, Martin Kuhlwilm, Peter Frandsen, Vitor C. Sousa, Tariq Desai, Javier Prado-Martinez, Jessica Hernandez-Rodriguez, et al. 2016. "Chimpanzee Genomic Diversity Reveals Ancient Admixture with Bonobos." *Science* 354 (6311): 477–81.
- McManus, Kimberly F., Joanna L. Kelley, Shiya Song, Krishna R. Veeramah, August E. Woerner, Laurie S. Stevison, Oliver A. Ryder, et al. 2015. "Inference of Gorilla Demographic and Selective History from

- Whole-Genome Sequence Data." *Molecular Biology and Evolution* 32 (3): 600–612.
- Roy, Justin, Maryke Gray, Tara Stoinski, Martha M. Robbins, and Linda Vigilant. 2014. "Fine-Scale Genetic Structure Analyses Suggest Further Male than Female Dispersal in Mountain Gorillas." *BMC Ecology* 14 (July): 21.
- Tricou, Théo, Eric Tannier, and Damien M. de Vienne. 2022. "Ghost Lineages Can Invalidate or Even Reverse Findings Regarding Gene Flow." *PLoS Biology* 20 (9): e3001776.
- Vernot, Benjamin, Serena Tucci, Janet Kelso, Joshua G. Schraiber, Aaron B. Wolf, Rachel M. Gitterman, Michael Dannemann, et al. 2016. "Excavating Neandertal and Denisovan DNA from the Genomes of Melanesian Individuals." *Science* 352 (6282): 235–39.

Decision Letter, first revision:

23rd May 2023

Dear Dr. Kuhlwilm,

Thank you for submitting your revised manuscript "Ghost admixture in eastern gorillas" (NATECOLEVOL-221218245A). It has now been seen again by the original reviewers and their comments are below. The reviewers find that the paper has improved in revision, and therefore we'll be happy in principle to publish it in Nature Ecology & Evolution, pending minor revisions to satisfy the reviewers' final requests and to comply with our editorial and formatting guidelines.

[REDACTED]

Reviewer #1 (Remarks to the Author):

I have reviewed the changes and find the revisions satisfactory, because my major concerns about the demographic modelling have been properly addressed.

Reviewer #2 (Remarks to the Author):

The authors have done a great job of addressing all my concerns and the manuscript is much improved!

I am happy with the current state of the manuscript.

Reviewer #3 (Remarks to the Author):

The authors have successfully addressed my previous concerns in their revision. I recommend this manuscript for publication.

29Our ref: NATECOLEVOL-221218245A

26th May 2023

Dear Dr. Kuhlwilm,

Thank you for your patience as we've prepared the guidelines for final submission of your Nature Ecology & Evolution manuscript, "Ghost admixture in eastern gorillas" (NATECOLEVOL-221218245A). Please carefully follow the step-by-step instructions provided in the attached file, and add a response in each row of the table to indicate the changes that you have made. Please also check and comment on any additional marked-up edits we have proposed within the text. Ensuring that each point is addressed will help to ensure that your revised manuscript can be swiftly handed over to our production team.

****We would like to start working on your revised paper, with all of the requested files and forms, as soon as possible (preferably within two weeks). Please get in contact with us immediately if you anticipate it taking more than two weeks to submit these revised files.****

In recognition of the time and expertise our reviewers provide to Nature Ecology & Evolution's editorial process, we would like to formally acknowledge their contribution to the external peer review of your manuscript entitled "Ghost admixture in eastern gorillas". For those reviewers who give their assent, we will be publishing their names alongside the published article.

Nature Ecology & Evolution offers a Transparent Peer Review option for new original research manuscripts submitted after December 1st, 2019. As part of this initiative, we encourage our authors to support increased transparency into the peer review process by agreeing to have the reviewer comments, author rebuttal letters, and editorial decision letters published as a Supplementary item. When you submit your final files please clearly state in your cover letter whether or not you would like to participate in this initiative. Please note that failure to state your preference will result in delays in accepting your manuscript for publication.

Cover suggestions

30As you prepare your final files we encourage you to consider whether you have any images or illustrations that may be appropriate for use on the cover of Nature Ecology & Evolution.

Nature Ecology & Evolution has now transitioned to a unified Rights Collection system which will allow our Author Services team to quickly and easily collect the rights and permissions required to publish your work. Approximately 10 days after your paper is formally accepted, you will receive an email in providing you with a link to complete the grant of rights. If your paper is eligible for Open Access, our Author Services team will also be in touch regarding any additional information that may be required to arrange payment for your article.

Please note that *Nature Ecology & Evolution* is a Transformative Journal (TJ). Authors may publish their research with us through the traditional subscription access route or make their paper immediately open access through payment of an article-processing charge (APC). Authors will not be required to make a final decision about access to their article until it has been accepted. [Find out more about Transformative Journals](https://www.springernature.com/gp/open-research/transformative-journals)

Authors may need to take specific actions to achieve [compliance with funder and institutional open access mandates](https://www.springernature.com/gp/open-research/funding/policy-compliance-faqs). If your research is supported by a funder that requires immediate open access (e.g. according to [Plan S principles](https://www.springernature.com/gp/open-research/plan-s-compliance)) then you should select the gold OA route, and we will direct you to the compliant route where possible. For authors selecting the subscription publication route, the journal's standard licensing terms will need to be accepted, including [self-archiving-and-license-to-publish](https://www.nature.com/nature-portfolio/editorial-policies/self-archiving-and-license-to-publish). Those licensing terms will supersede any other terms that the author or any third party may assert apply to any version of the manuscript.

For information regarding our different publishing models please see our [page](https://www.springernature.com/gp/open-research/transformative-journals) Transformative Journals page. If you have any questions about costs, Open Access requirements, or our legal forms, please contact ASJournals@springernature.com.

[REDACTED]

[REDACTED]

Reviewer #1:

Remarks to the Author:

I have reviewed the changes and find the revisions satisfactory, because my major concerns about the demographic modelling have been properly addressed.

Reviewer #2:

Remarks to the Author:

The authors have done a great job of addressing all my concerns and the manuscript is much improved!

I am happy with the current state of the manuscript.

Reviewer #3:

Remarks to the Author:

The authors have successfully addressed my previous concerns in their revision. I recommend this manuscript for publication.

Final Decision Letter:

30th June 2023

Dear Professor Kuhlwilm,

32We are pleased to inform you that your Article entitled "Ghost admixture in eastern gorillas", has now been accepted for publication in Nature Ecology & Evolution.

Over the next few weeks, your paper will be copyedited to ensure that it conforms to Nature Ecology and Evolution style. Once your paper is typeset, you will receive an email with a link to choose the appropriate publishing options for your paper and our Author Services team will be in touch regarding any additional information that may be required

You will not receive your proofs until the publishing agreement has been received through our system

Due to the importance of these deadlines, we ask you please us know now whether you will be difficult to contact over the next month. If this is the case, we ask you provide us with the contact information (email, phone and fax) of someone who will be able to check the proofs on your behalf, and who will be available to address any last-minute problems . Once your paper has been scheduled for online publication, the Nature press office will be in touch to confirm the details.

Acceptance of your manuscript is conditional on all authors' agreement with our publication policies (see www.nature.com/authors/policies/index.html). In particular your manuscript must not be published elsewhere and there must be no announcement of the work to any media outlet until the publication date (the day on which it is uploaded onto our web site).

Please note that *Nature Ecology & Evolution* is a Transformative Journal (TJ). Authors may publish their research with us through the traditional subscription access route or make their paper immediately open access through payment of an article-processing charge (APC). Authors will not be required to make a final decision about access to their article until it has been accepted. [Find out more about Transformative Journals](https://www.springernature.com/gp/open-research/transformative-journals)

Authors may need to take specific actions to achieve [compliance](https://www.springernature.com/gp/open-research/funding/policy-compliance-faqs) with funder and institutional open access mandates. If your research is supported by a funder that requires immediate open access (e.g. according to [Plan S principles](https://www.springernature.com/gp/open-research/plan-s-compliance)) then you should select the gold OA route, and we will direct you to the compliant route where possible. For authors selecting the subscription publication route, the journal's standard licensing terms will need to be accepted, including [self-archiving-and-license-to-publish](https://www.nature.com/nature-portfolio/editorial-policies/self-archiving-and-license-to-publish). Those licensing terms will supersede any other terms that the author or any third party may assert apply to any version of the manuscript.

In approximately 10 business days you will receive an email with a link to choose the appropriate

33publishing options for your paper and our Author Services team will be in touch regarding any additional information that may be required.

We welcome the submission of potential cover material (including a short caption of around 40 words) related to your manuscript; suggestions should be sent to Nature Ecology & Evolution as electronic files (the image should be 300 dpi at 210 x 297 mm in either TIFF or JPEG format). Please note that such pictures should be selected more for their aesthetic appeal than for their scientific content, and that colour images work better than black and white or grayscale images. Please do not try to design a cover with the Nature Ecology & Evolution logo etc., and please do not submit composites of images related to your work. I am sure you will understand that we cannot make any promise as to whether any of your suggestions might be selected for the cover of the journal.

You can generate the link yourself when you receive your article DOI by entering it here: <http://authors.springernature.com/share>.

Yours sincerely,

Luíseach

Luíseach Nic Eoin, DPhil
Senior Editor
Nature Ecology and Evolution

My pronouns: she/her

34P.S. Click on the following link if you would like to recommend Nature Ecology & Evolution to your librarian <http://www.nature.com/subscriptions/recommend.html#forms>

** Visit the Springer Nature Editorial and Publishing website at http://editorial-jobs.springernature.com?utm_source=ejp_NEcoE_email&utm_medium=ejp_NEcoE_email&utm_campaign=ejp_NEcoE for more information about our career opportunities. If you have any questions please click [here](mailto:editorial.publishing.jobs@springernature.com).**